# Towards Deep Radar Perception for Autonomous Driving: Datasets, Methods, and Challenges

**DOI:** 10.3390/s22114208

**Published:** 2022-05-31

**Authors:** Yi Zhou, Lulu Liu, Haocheng Zhao, Miguel López-Benítez, Limin Yu, Yutao Yue

**Affiliations:** 1Institute of Deep Perception Technology (JITRI), Wuxi 214000, China; zhouyi1023@tju.edu.cn (Y.Z.); lulu.liu21@student.xjtlu.edu.cn (L.L.); haocheng.zhao19@student.xjtlu.edu.cn (H.Z.); 2Department of Electrical and Electronic Engineering, School of Advanced Technology, Xi’an Jiaotong-Liverpool University, Suzhou 215123, China; limin.yu@xjtlu.edu.cn; 3XJTLU-JITRI Academy of Industrial Technology, Xi’an Jiaotong-Liverpool University, Suzhou 215123, China; 4Department of Electrical Engineering and Electronics, University of Liverpool, Liverpool L69 3GJ, UK; mlopben@liverpool.ac.uk; 5Department of Mathematical Sciences, School of Science, Xi’an Jiaotong-Liverpool University, Suzhou 215123, China; 6Department of Mathematical Sciences, University of Liverpool, Liverpool L69 7ZX, UK; 7ARIES Research Centre, Antonio de Nebrija University, 28040 Madrid, Spain

**Keywords:** automotive radars, radar signal processing, object detection, multi-sensor fusion, deep learning, autonomous driving

## Abstract

With recent developments, the performance of automotive radar has improved significantly. The next generation of 4D radar can achieve imaging capability in the form of high-resolution point clouds. In this context, we believe that the era of deep learning for radar perception has arrived. However, studies on radar deep learning are spread across different tasks, and a holistic overview is lacking. This review paper attempts to provide a big picture of the deep radar perception stack, including signal processing, datasets, labelling, data augmentation, and downstream tasks such as depth and velocity estimation, object detection, and sensor fusion. For these tasks, we focus on explaining how the network structure is adapted to radar domain knowledge. In particular, we summarise three overlooked challenges in deep radar perception, including multi-path effects, uncertainty problems, and adverse weather effects, and present some attempts to solve them.

## 1. Introduction

As autonomous driving technology progresses from the demonstration stage to the landing stage, it puts forward higher requirements for perception ability. Mainstream autonomous driving systems rely on the fusion of cameras and LiDARs for perception. Although millimetre wave radar has been widely used in mass-produced cars for active safety functions such as automatic emergency braking (AEB) and forward collision warning (FCW), it is overlooked in autonomous driving. Recently, Tesla announced the removal of radar sensors from its semi-autonomous driving system Autopilot. In the CVPR 2021 workshop [1], Tesla’s director of AI, Andrej Karpathy, explained their reason by presenting three typical scenarios for radar malfunctions, including lost tracking due to significant deceleration of the front vehicle, false slow down under bridges, and missed detection of a stationary vehicle parked on the side of the main road. In the first case, radar’s close field detection ability is related to sidelobes. Conventional radars with a limited number of channels are not good at sidelobe compression. The second case is caused by the fact that conventional radar cannot measure height information and, therefore, confuses the bridge overhead with static objects on the road. The reason for the third case is that conventional radar has a too low angular resolution to capture the shape of a static vehicle. All these challenges can be solved with next-generation high-resolution radar.

As a ranging sensor, radar is usually compared to LiDAR. A typical 77 Ghz automotive radar has a wavelength of 3.9 mm, while automotive LiDARs have a much smaller wavelength of 905 nm or 1550 nm. For a small-aperture radar, most of the reflected signal is not received by the radar sensor because of the specular reflection. Another problem with a small aperture is the low angular resolution, so that two close objects cannot be separated effectively. These two features make the radar point cloud much sparser than the LiDAR point cloud. Conventional automotive radars have a low resolution in elevation and, therefore, return a two-dimensional point cloud. The next generation of high-resolution radar achieves higher angular resolution in both azimuth and elevation. Because it can measure 3D position and Doppler velocity, it is always referred to as a 4D radar in the marketplace. In Table 1, typical types of radars and LiDARs are compared. We can find that conventional long-range radars have a low angular resolution in horizontal view and no resolution in vertical view. In contrast, 4D radar can achieve an angular resolution of about 1° in both horizontal and vertical views. Therefore, the classification of static objects is no longer a limitation for 4D radar. Although 4D radar has a much higher angular resolution, as shown in Figure 1, the radar point cloud is still much sparser than the 16-beam LiDAR point cloud. However, radar can measure Doppler velocity and radar cross-section (RCS), which is expected to better help classify road users. In addition, 4D radar has the advantages of a long detection range (up to 300 m), all-weather operation, low power consumption, and low cost. Therefore, we believe that radar is a good complement to LiDAR and vision. The fusion of these sensors enables all-weather, long-range environment perception.

In recent years, with the trend of open-source, more and more datasets, models, and toolboxes have been released. According to our statistics, 10 radar datasets were released in 2021 and 2022. Along with these datasets, some seminal papers are proposed to leverage deep learning in radar perception. However, due to the limited sensing capability of conventional radar, the performance of these methods is far from good enough. Since the introduction of 4D radar, we believe that the era of deep radar perception has arrived. With the power of deep learning, we can design a highly reliable perception system based on the fusion of radar and other modalities.

The application of deep learning in radar perception has drawn extensive attention from autonomous driving researchers. In the past two years, a number of review papers [3,4,5,6,7,8] have been published in this field. Zhou et al. [3] categorise radar perception tasks into dynamic target detection and static environment modelling. They also provide brief introductions to radar-based detection, tracking, and localisation. Abdu et al. [4] summarise the deep learning models for radar perception based on different radar representations, including occupancy grid maps, range–Doppler–azimuth maps, micro-Doppler spectrograms, and point clouds. They also introduce approaches for radar and camera fusion based on the classical taxonomy of the data-level, feature-level, and decision-level. Scheiner et al. [5] discuss the information sparsity problem and labelling challenge in learning-based radar perception. Three strategies are recommended to increase radar data density, including the use of pre-CFAR data, the use of high-resolution radar sensors, and the use of polarisation information. In this paper, we differ from other review papers in three aspects: Firstly, we provide a detailed summary and description of the publicly available radar datasets, which is very useful for developing learning-based methods. Secondly, this review does not focus on specific tasks, but aims to present a big picture of the radar perception framework, as illustrated in Figure 2. Thirdly, rather than simply presenting the network structure, we focus on explaining why these modules work from the perspective of radar domain knowledge.

In this article, we systematically review the recent advancements in deep radar perception. In Section 2, we introduce the radar signal processing pipeline and different radar data representations. In Section 3, we summarise the publicly available radar datasets (Section 3.1) for autonomous driving, as well as the calibration (Section 3.2), labelling (Section 3.3), and data augmentation techniques (Section 3.4). In the following sections, we introduce different radar perception tasks, including radar depth completion (Section 4.1), radar full-velocity estimation (Section 4.2), and radar object detection (point-cloud-based in Section 5.2 and pre-CFAR-based in Section 5.3). In Section 6, we classify sensor fusion frameworks into four categories: input fusion (Section 6.1), ROI fusion (Section 6.2), feature map fusion (Section 6.3), and decision fusion (Section 6.4). Next, we discuss three challenges toward reliable radar detection, including ghost objects (Section 7.1), uncertainty problems (Section 7.2), and adverse weather effects (Section 7.3). Finally, we propose several interesting research directions in Section 8.

## 2. Radar Signal Processing Fundamentals

Knowledge of radar signal processing is essential for the development of a deep radar perception system. Different radar devices vary in their sensing capabilities. It is important to leverage radar domain knowledge to understand the performance boundary, find key scenarios, and solve critical problems. This section outlines the classical signal processing pipeline for automotive radar applications.

### 2.1. FMCW Radar Signal Processing

Off-the-shelf automotive radars operate with a sequence of linear frequency-modulated continuous-wave (FMCW) signals to simultaneously measure range, angle, and velocity. According to regulations, automotive radar is allowed to use two frequency bands in millimetre waves: 24 GHz (24–24.25 GHz) and 77 GHz (77–79 GHz). There is a trend towards 77 GHz due to its larger bandwidth (76–77 GHz for long-range and 77–81 GHz for short-range), higher Doppler resolution, and smaller antennas [9]. As shown in Figure 3, the FMCW signal is characterised by the start frequency (also known as the carrier frequency) $f_{c}$, the sweep bandwidth *B*, the chirp duration $T_{c}$, and the slope $S=B/T_{c}$. During one chirp duration, the frequency increases linearly from $f_{c}$ to $f_{c}+B$ with a slope of S. One FMCW waveform is referred to as a chirp, and radar transmits a frame of $N_{c}$ chirps equally spaced by chirp cycle time $T_{c}$. The total time $T_{f}=N_{c}T_{c}$ is called the frame time, also known as the time on target (TOT). In order to avoid the need for high-speed sampling, a frequency mixer combines the received signal with the transmitted signal to produce two signals with sum frequency $f_{T}\left(t\right)+f_{R}\left(t\right)$ and difference frequency $f_{T}\left(t\right)-f_{R}\left(t\right)$. Then, a low-pass filter is used to filter out the sum frequency component and obtain the intermediate frequency (IF) signal. In this way, FMCW radar can achieve GHz performance with only MHz sampling. In practice, a quadrature mixer is used to improve the noise figure [10], resulting in a complex exponential IF signal as
(1)$$x_{\mathrm{IF}}\left(t\right)=Ae^{j(2\pi f_{\mathrm{IF}}t+\phi_{\mathrm{IF}})}$$
where *A* is the amplitude, $f_{\mathrm{IF}}=f_{T}\left(t\right)-f_{R}\left(t\right)$ is referred to as the beat frequency, and $\phi_{\mathrm{IF}}$ is the phase of the IF signal. Next, the IF signal is sampled $N_{s}$ times by an ADC converter, resulting in a discrete-time complex signal. Multiple frames of chirp signals are assembled into a two-dimensional matrix. As shown in Figure 3, the dimension of the sampling points within a chirp is referred to as fast time, and the dimension of the chirp index within one frame is referred to as slow time. Assuming one object moving with speed *v* at distance *r*, the frequency and phase of the IF signal are given by
(2)$$f_{\mathrm{IF}}=\frac{2S(r+vT_{c})}{c},\phi_{\mathrm{IF}}=\frac{4\pi (r+vT_{c})}{\lambda }$$
where $\lambda =c/f_{c}$ is the wavelength of the chirp signal. From (Equation 2), we can find that the range and Doppler velocity are coupled. Under the following assumptions: 1. the range variations in slow time caused by target motion can be neglected due to the short frame time; 2. the Doppler frequency in fast time can be neglected compared to the beat frequency by utilising a wideband waveform. Then, range and Doppler can be decoupled. Range can be estimated from the beat frequency as $r=cf_{\mathrm{IF}}/2S$, and Doppler velocity can be estimated from the phase shift between two chirps as $v=\Delta \phi \lambda /4\pi T_{c}$. Next, a range DFT is applied in the fast-time dimension to resolve the frequency change, followed by a Doppler DFT in the slow-time dimension to resolve the phase change. As a result, we obtain a 2D complex-valued data matrix called the Range–Doppler (RD) map. In practice, a window function is applied before DFT to reduce sidelobes. The range and the Doppler velocity of a cell in the RD map are given by
(3)$$r_{k}=k\frac{c}{2B_{\mathrm{IF}}},v_{l}=l\frac{\lambda }{2T_{f}}$$
where *k* and *l* denote the indexes of DFT, $B_{\mathrm{IF}}$ is the IF bandwidth, and $T_{f}$ is the frame time. In practice, FFT is applied due to its computational efficiency. Accordingly, the sequence will be zero-padded to the nearest power of 2 if necessary.

Angle information can be obtained using more than one receive or transmit channel. Single-input multiple-output (SIMO) radars utilise a single transmit (Tx) and multiple receive (Rx) antennas for angle estimation. Suppose one object is located in direction θ. Similar to Doppler processing, the induced frequency change between two adjacent receive antennas can be neglected, while the induced phase change can be used for calculating the direction of the angle. This phase change is given by Δϕ=2πdsinθ/λ, where *d* is the inter-antenna spacing. To achieve the maximum unambiguous angle, the spacing can be set to λ/2. Then, a third FFT can be applied to the receive antenna dimension. For conventional radar with a small number of Rx antennas, the sequence is often padded with $N_{\mathrm{FFT}}-N_{Rx}$ zeros to achieve a smooth display of the spectrum. The angle at index η is given by
(4)$$\theta_{\eta }=\arcsin\frac{\eta \lambda }{N_{\mathrm{FFT}}}$$

The angular resolution of a SIMO radar depends on the number of Rx antennas. The maximum number of Rx antennas is limited by the additional cost of signal processing chains on the device [11]. Multiple-input multiple-output (MIMO) radar operates with multiple channels in both Tx and Rx. As illustrated in Figure 4a, a MIMO radar with $N_{\mathrm{Tx}}$ Tx antennas and $N_{\mathrm{Rx}}$ Rx antennas can synthesise a virtual array with $N_{\mathrm{Tx}}N_{\mathrm{Rx}}$ channels. In order to separate the transmit signals at the receiver side, the signals from different Tx antennas should be orthogonal. There are multiple ways to realise waveform orthogonality, such as time-division multiplexing (TDM), frequency-division multiplexing (FDM), and Doppler-division multiplexing (DDM) [12,13]. TDM is widely used for its simplicity. In this mode, different Tx antennas transmit chirp signals in turns, as shown in Figure 4b. Therefore, at the receiver side, different Tx waveforms can be easily separated in the time domain. An additional phase shift compensation [14] is required to compensate for the motion of detections during the Tx switching time. Another shortcoming of TDM is the reduced detection range due to the loss of transmitting power. DDM is also supported by many radar devices. As shown in Figure 4c, DDM transmits all Tx waveforms simultaneously and separates them in the Doppler domain. In order to realise waveform orthogonality, for the *k*-th transmitter, a Doppler shift is added to adjacent chirps as
(5)$$\omega_{k}=\frac{2\pi (k-1)}{N}$$
where *N* is usually selected as the number of Tx antennas $N_{\mathrm{Tx}}$. One drawback of DDM is that its unambiguous Doppler velocity is reduced to $\frac{1}{N}$ of the original one. Empty-band DDM [15] can achieve more robust velocity disambiguation by introducing several empty Doppler sub-bands. Some example codes are provided in the RADIal dataset [16]. After decoupling the received signals, we can obtain a 3D tensor by stacking RD maps with respect to Tx–Rx pairs. Then, the DOA can be estimated through the angle FFT along the virtual receiver dimension. Some super-resolution methods [17], such as Capon, MUSIC, and ESPRIT, can be applied to improve angular resolution. The resulting 3D tensor is referred to as the range–azimuth–Doppler (RAD) tensor or radar tensor.

In the radar detection pipeline, RD maps are integrated coherently along the virtual receiver dimension to increase the SNR. Then, a constant false alarm rate (CFAR) detector [18] is applied to detect peaks in the RD map. Finally, the DOA estimation method is applied for angle estimation. The output is a point cloud with measurements of range, Doppler, and angle. For conventional radars, only the azimuth angle is resolved, while 4D radars output both azimuth and elevation angles. Since radar is usually used in safety-critical applications, a lower CFAR threshold (≤10 dB) is set to achieve high recall. The accuracy of detection is affected by road clutter, interferences, and multi-path effects in complex environments. Therefore, additional spatial–temporal filtering is required to improve accuracy. DBSCAN [19] is used to cluster radar detections into object-level targets. Clusters with few detections are considered as outliers and, thus, be removed. Further, temporal filtering, such as Kalman filtering, is used to filter out outliers and interpolate missed detections.

### 2.2. Radar Performances

The performance of automotive radar can be evaluated in terms of maximum range, maximum Doppler velocity, and field of view (FoV). Equations for these attributes are summarised in Table 2. According to the radar equation, the theoretical maximum detection range is given by
(6)$$R_{\max}=\sqrt[{4}]{\frac{P_{t}G^{2}\lambda^{2}\sigma }{{\left(4\pi \right)}^{3}P_{\min}}}$$
where $P_{t}$ is the transmit power, $P_{\min}$ is the minimum detectable signal or receiver sensitivity, λ is the transmit wavelength, σ is the target RCS, and *G* is the antenna gain. The wavelength is 3.9 mm for automotive 77 GHz radar. The target RCS is a measure of the ability to reflect radar signals back to the radar receiver. It is a statistical quantity that varies with the viewing angle and the target material. According to the test results [20], smaller objects, such as pedestrians and bikes, have an average RCS value of around 2–3 dBsm, whereas normal vehicles have an average RCS value of around 10 dBsm and large vehicles of around 20 dBsm. The other parameters, such as transmit power, minimum detectable signal, and antenna gain are design parameters aimed at meeting product requirements, as well as regulations. Some typical values for these parameters are summarised in Table 3. In practice, the maximum range is limited by the supported IF bandwidth $B_{\mathrm{IF}}$ and ADC sampling frequency. The maximum unambiguous velocity is inversely proportional to the chirp duration $T_{c}$. For MIMO radar, the maximum unambiguous angle is dependent on the spacing of antennas *d*. The theoretical maximum FoV is 180° if d=λ/2. In practice, the FoV is determined by the antenna gain pattern. Another important characteristic is resolution, i.e., the ability to separate two close targets with respect to range, velocity, and angle. As shown in Table 2, high-range resolution requires a large sweep bandwidth *B*. High Doppler resolution requires a long integration time, i.e., the frame time $N_{c}T_{c}$. The angular resolution depends on the number of virtual receivers $N_{R}$, the object angle θ, and the inter-antenna spacing *d*. For the case of d=λ/2 and θ=0°, angular resolution is in a simple form of $2/N_{R}$. From the perspective of antenna theory, angular resolution can also be featured by the half-power beamwidth, i.e., the 3-dB beamwidth [13], which is a function of the array aperture *D*.

In practice, different types of automotive radar are designed for different scenarios. Long-range radar (LRR) achieves a long detection range and a high angular resolution at the cost of a smaller FoV. Short-range radar (SRR) uses MIMO techniques to achieve a high angular resolution and large FoV. In addition, different chirp configurations [21] are used for different applications. For example, long-range radar needs to detect fast-moving vehicles at distances and, therefore, utilises a small ramp slope for long-distance detection, a long chirp integration time to increase the SNR, a small chirp duration to increase the maximum velocity, and a short chirp duration for high-velocity resolution [22]. Short-range radar needs to detect vulnerable road users (VRUs) close to the vehicle and, therefore, utilises a higher sweep bandwidth for high-range resolution at the cost of a short range. Multi-mode radar [21] can work in different modes simultaneously by sending chirps that are switched sequentially with different configurations.

### 2.3. Open-Source Radar Toolbox

Commercial off-the-shelf radar products can only output point clouds. They can be configured to output either raw point clouds, sometimes referred to as radar detections, or clustered objects with tracked IDs. The signal processing algorithm inside it is a black box and cannot be modified. Alternatively, TI mm-wave radars have been widely used in academic research because of their public nature and flexibility. They support configurable chirps [21] and different MIMO modes [11] to adapt to different tasks. TI also provides a mmWave studio, which provides GUIs for radar setup, data capturing, signal processing, and visualisation. In addition, there are some open-source radar signal processing toolboxes for TI devices, for example RaDICaL SDK [23,24], PyRapid [25], OpenRadar [26], and Pymmw [27]. These toolboxes enable researchers to build their own datasets using TI devices. While there is a growing number of public radar datasets, most of them provide limited information about the radar configurations they use. This makes it difficult to make a fair comparison between algorithms trained on different datasets. The open radar initiative [28] provides a guideline for radar configuration and encourages researchers to expand this dataset by using the radar device with the same configuration.

## 3. Datasets, Labelling, and Augmentation

Data play a key role in the learning-based approaches. In the past, radar algorithms were always evaluated on private datasets. Recently, with the trend towards open-source, many radar datasets have become publicly available. In this section, we summarise these radar datasets with respect to their data representations, tasks, scenarios, and annotation types. To motivate readers to build their own datasets, we also introduce extrinsic calibration and cross-modality labelling techniques. We further investigate data augmentation methods and the potential use of synthetic radar data to improve data diversity.

### 3.1. Radar Datasets

Different radar datasets use different types of radar. We can classify radar sensors into low resolution (LR) and high resolution (HR). There are different technical routes to achieve high resolution, such as polarimetric radar [29], cooperative radars [30], multi-chip cascaded MIMO radar [13], synthetic aperture radar (SAR) [31], and spinning radar [32]. Most off-the-shelf radars can output a point cloud with range, azimuth angle, Doppler velocity, and RCS. Next-generation 4D radar can also measure elevation angle. Some radar prototypes can be configured to output radar raw data, including ADC data, RA/RD maps, and RAD tensors.

The role of radar in autonomous driving can be divided into localisation and detection. Although this paper focuses on radar detection, we also introduce the localisation datasets in this section. Since these datasets usually provide synchronised LiDAR and images along with radar data, it is possible to annotate them for detection purpose, as done in [33]. There are various levels of label granularity for radar data. For radar point clouds, it is possible to provide 2D bounding boxes, 3D bounding boxes, or pointwise annotations. The 2D bounding boxes are labelled in bird’s eye view (BEV) and with orientation information; hence, they are sometimes referred to as pseudo-3D boxes. The 3D bounding boxes further capture height information and pitch angle. If properly annotated, pointwise annotation can provide semantic information at a finer granularity than bounding boxes. In fact, radar detections within the bounding box could also be ghost detection or clutter. Therefore, pointwise annotation is a better way to capture the noisy nature of the radar point cloud. Similarly, radar pre-CFAR data, including RA/RD maps and RAD tensors, are also annotated pointwise. Some works dilate the annotated points into a dense mask or a bounding box. However, these dilated patches do not necessarily reflect the shape information. Some techniques for precise dilation will be introduced later in Section 4.

There are some large-scale datasets for autonomous driving that include off-the-shelf 2D radars in their sensor suites. NuScenes [34] is the most popular dataset for its large-scale and diverse scenarios. The capturing vehicle is equipped with a 32-beam LiDAR, 6 cameras, 5 long-range multi-mode radars, and a GPS/IMU system. It provides 3D annotations of 23 classes of road users in 1000 scenes, with a total of 1.3 million frames. However, this dataset is not a good choice for studying the role of radar in perception, because its radar point clouds are too sparse. The PixSet dataset [35] also aims at 3D object detection. The vehicle is equipped with a colocated sensor platform consisting of a solid-state LiDAR, a 64-beam LiDAR, a TI AWR1843 radar, and a GPS/IMU system. The FoVs of different modalities are largely overlapped and, hence, are well suited for evaluating sensor fusion algorithms. The RadarScenes dataset [36] is a diverse large-scale dataset for instance segmentation of radar point clouds. It uses four 77 GHz radars with overlapping FoVs in the front of the vehicle. Each radar is in middle-range mode with a maximum range of 100 m and a 60° FoV. Compared to the NuScenes dataset, its radar point clouds are much denser. The dataset contains 100 km of driving in 158 different scenarios. It provides both pointwise annotations and track IDs for 11 classes of moving road users. All points with zero velocity are labelled as static. Pointillism [37] leverages a multi-radar setup to improve the resolution. Two TI IWR1443 radars were placed at the front of the car, facing forward, at a distance of 1.5 m. The aim is to study the effect of coherently integrating point clouds from two radar sensors. In order provide the ground truth of radar point clouds, the sensor suite also include a 16-beam LiDAR and a camera with overlapping FoVs. The dataset contains 54K synchronised frames for five typical driving scenarios under different weather conditions. It also provides 3D box annotations of vehicles. The Zendar dataset [38] is a high-resolution radar dataset that uses SAR for moving vehicle detection. It provides time-synchronised images, radar ADC data, 2D SAR point clouds, and projected LiDAR point clouds in BEV. Pointwise annotations of moving vehicles are applied to the SAR point cloud. It also provides an SDK for converting raw ADC data to RD maps and visualisation.

Robust perception under adverse weather is a popular research topic for safe autonomous driving. Although there are some recently published datasets for adverse weather [39,40,41,42], only a few include radar in their sensor suite. The Dense dataset [43] focused on evaluating multi-modal fusion algorithms under adverse weather. In addition to LiDAR and a stereo camera, it is also equipped with several all-weather sensors, including one frontal long-range radar, one gated camera working on the NIR band, one FIR camera, and one weather station sensor. The data are captured in various natural weather conditions, including rain, snow, light fog, and heavy fog, as well as in a controlled lab environment in a fog chamber. However, the dataset only provides sparse radar targets with limited FoV and poor resolution. The RADIATE dataset [44] focuses in particular on leveraging radar in adverse weather. The data-collection car is equipped with a camera, a LiDAR, and a spinning radar. The datasets are captured under different weather, such as sun, night, rain, fog, and snow. It provides annotations for 2D object detection, object tracking, and SLAM.

Several datasets utilise radar sensors in short-range (SR) or ultra-short-range (USR) mode for high-resolution close-field imaging. In this mode, close objects will occupy several cells in both the range and Doppler dimension (because of the micro-Doppler motion). To fully utilise these spatially spread range and Doppler signatures, annotations are made directly on RA maps or RAD tensors. The CARRADA dataset [45] uses TI AWR1843BOOST radar in short-range mode, with a max distance of 50 m. It provides real-valued RA maps, RD maps, and unannotated RAD tensors, as well as synchronised images, for training neural networks. In both RA and RD maps, objects are annotated at the point level with categories of pedestrian, car, or cyclist. In addition, the dilated segmentation mask and the bounding box around the cluster are also provided. The data are collected on an empty test track with at most two moving objects in the FoV. The RADDet dataset [46] also uses TI AWR1843BOOST radar with a max distance of 50 m, as well as a stereo camera. It provides 3D bounding boxes for complex-valued RAD tensors and 2D bounding boxes for RA maps projected in Cartesian view. The data are captured using a tripod located on the sidewalks and facing the main roads. Therefore, its scenario is much more complex than the CARRADA dataset. The CRUW dataset [47] uses a TI AWR1843 radar and a stereo camera for object detection. It adopts a different signal processing pipeline, which directly outputs the RA map using range FFT and angle FFT. Then, the object-level pointwise annotation is applied to complex-valued RA maps. A probabilistic camera-0radar fusion approach is used to improve annotation quality. The dataset contains 3.5 h and 400 K frames of camera–radar data in different driving scenarios, including parking lot, campus road, city street, and highway. The RaDICaL dataset [48] uses TI IWR1443BOOST radar in multiple configurations for different scenarios, including indoor, parking lot, highway, and single human walking. It records radar ADC data together with the RGB-D images and IMU data using ROS. It also provides a signal processing SDK [23] to process and annotate radar data. The Ghent VRU dataset [49] collects radar data specifically for VRU detection. The sensor suite includes a TI AWR1243 radar, a camera, and a 16-beam LiDAR. The data are recorded by a vehicle driving on public roads in a crowded European city centre. It provides radar RAD tensors with segmentation mask annotations for VRUs. To compensate for range-dependent power, many datasets apply logarithmic scaling or normalisation to the pre-CFAR data as the default. The CARRADA dataset [45] and RADDet dataset [46] apply logarithmic scaling to their radar data. The normalisation can be applied in different ways, including local power normalising in the Ghent VRU dataset [49], min–max scaling in the CRUW dataset [47], and Z-score standardisation in the RADDet dataset [46]. Here, we only summarize some operations that are explicitly mentioned. Further checks are needed when benchmarking the algorithm using different datasets.

As 4D radar is just entering the market, only a few public datasets are available. The Astyx dataset [2] is the first publicly available 4D radar dataset. The sensors include a 16-beam LiDAR, a camera, and an Astyx 6455 HiRes 4D radar. It provides 3D bounding box annotations of seven classes of road users. Each object also features four levels of occlusion and three levels of uncertainty. The dataset is very small, with only 500 annotated frames of short clips, each clip containing less than 10 frames. The class distribution is very imbalanced, with over 90% of the annotated objects being car. The View-of-Delft (VoD) dataset [50] is a recently published 4D radar dataset especially focused on the detection of VRUs. The sensor suite includes a ZF FRGen 21 4D radar, a 64-beam LiDAR, and a stereo camera. It provides 8693 annotated frames with 3D bounding boxes and tracking IDs. Each object is also annotated with two levels of occlusion and four types of activity attributes (stopped, moving, parked, pushed, sitting). The data were collected in campus, suburb, and old-town locations, with a preference for scenarios containing VRUs. It provides fine-grained annotations of vehicles, trucks, and 10 classes of VRUs. Different classes are equally distributed (21.6% pedestrians, 8.8% cyclists, and 21.9% cars). The RADIal dataset [51] is a 4D radar dataset for vehicle detection and open space segmentation. The sensors include camera, LiDAR, 4D radar, GPS. and vehicle’s CAN traces. The 4D radar is a 12Tx 16Rx cascaded radar. A key feature is that they also provide radar raw ADC data, which makes it possible to explore the potential of neural networks in the signal processing stage. This dataset is comparable in size to the VoD dataset, with 8252 annotated frames captured in city streets, highways, and countryside roads. Two kinds of annotations are provided, vehicle annotations and open space segmentation masks in BEV. The vehicle annotations are in the format of 2D bounding boxes for image- and object-level points for LiDAR and radar. Although they do not provide bounding box annotations for the radar point clouds, it is possible for researchers to annotate them on their own, given the LiDAR point clouds and images. TJ4DRadSet [52] is a 4D dataset for 3D detection and tracking. The sensor suite includes a 32-beam LiDAR, a camera, a high-performance 4D radar (Oculii Eagle), and a GNSS. By utilising Oculii’s virtual aperture imaging technique, this 4D radar can output a much denser point cloud than others. It has a maximum detection range of 400 m and an angular resolution of less than 1° in both azimuth and elevation. The data are captured in a wide range of road conditions in urban driving. The dataset contains a total of 40K frames of synchronised data, where 7757 frames of them are annotated with 3D bounding boxes and track IDs.

Radar can also be utilised for localisation. Compared to camera and LiDAR, radar has the advantages of a long detection range and robustness to occlusions. Millimetre waves can penetrate certain non-metallic objects, such as glass, Polywood, and clay bricks [32] and are less affected by dust, smoke, fog, rain, snow, and ambient lighting conditions [32]. Therefore, radar has great potential for mapping and localisation in adverse weather. The Oxford radar robotcar dataset [53] is the most popular dataset for radar SLAM. The test car is equipped with a rich set of sensors, including an FMCW spinning radar, two 32-beam LiDARs, a stereo camera, three monocular cameras, two 2D LiDARs, and a GPS/IMU system. The spinning radar can provide a 360° high-resolution intensity map of surrounding environments. However, it has no Doppler information and is rarely used in production cars due to the high price. The Mulran dataset [54] focuses on range-sensor-based place recognition. It uses a spinning radar and a 64-beam LiDAR to capture the surrounding environment. The recorded data are temporally (monthly revisits) and structurally (multi-city) diverse. The Borea dataset [55] aims at studying the effect of seasonal variation on long-term localisation. The sensor suite includes a spinning radar, a camera, a GPS/IMU system, and a 128-beam LiDAR. The data were collected by driving a repeated route over one year, thus capturing seasonal variations and adverse weather conditions. It provides a pose ground truth for the localisation task, as well as 3D bounding box annotations for object detection in sunny weather. Similar to the Borea dataset, the EU long-term dataset [56] aims at localisation in highly dynamic environments and long-term autonomy. Its sensor suite includes two stereo cameras, two 32-beam LiDARs, two fisheye cameras, a four-beam LiDAR, a 77 GHz long-range radar, and a 2D LiDAR facing the road. The Endeavour dataset [57] adopts five multi-mode radars to cover the 360° surrounding environment. It is also equipped with LiDARs and RTK-GPS to provide the ground truth for radar odometry. The ColoRadar dataset [58] utilises a compact moving sensor rig, which consists of a 64-beam LiDAR, a TI AWR2243 cascaded 4D radar, a TI AWR1843 radar, and an IMU. Three levels of radar data representation are provided, including raw ADC samples, range–azimuth–elevation–Doppler (RAED) tensors from the 4D radar, and point clouds from the single-chip radar. The data are gathered in a variety of scenarios, including highly diverse indoor environments, outdoor environments, and an underground mine.

There are also some radar datasets designed for specific tasks. PREVENTION [59] focuses on predicting inter-vehicle interactions. The data-collection car is equipped with one frontal long-range radar, two corner radars, one 32-beam LiDAR, and two cameras. It provides annotations of 2D bounding boxes, lane change behaviours, and trajectories. SCORP [60] is a radar dataset for open space segmentation in parking scenarios. It provides three kinds of radar data, including the RAD tensor, RA map, and BEV map. The Radar Ghost dataset [61] aims at studying the effect of multi-path propagation in autonomous driving. It provides pointwise annotations of real targets and four types of ghost targets.

We summarize these datasets in Table 4. As two popular research directions, radar pre-CFAR datasets and 4D radar datasets are listed independently. With a focus on radar, we specify radar information, including type, data format, maximum range, and whether it comes with Doppler velocity. In addition, we provide information on other sensor modalities, scenarios, weathers, and size, so that researchers can select appropriate datasets for their tasks. We also maintain a website for browsing each reference and related data processing codes at [62].

### 3.2. Extrinsic Calibration

Multi-sensor extrinsic calibration requires calibration targets to be observed simultaneously by different modalities. The trihedral corner reflector is widely used for radar calibration because of its high RCS. Multiple reflectors are usually placed outdoors to avoid multi-path propagation. The difficulty lies in making the calibration target visible to both radar and other sensors. El Natour et al. [63] built a calibration facility by placing one Luneburg lens and seven trihedral corner reflectors with known inter-distances. To make the reflectors visually detectable, they painted each surface with different colours. Peršić et al. [64] designed a compact calibration target that can be simultaneously detected by the camera, radar, and LiDAR. As shown in Figure 5a, they placed a triangle-shaped chequerboard pattern in front of a trihedral corner reflector. The chequerboard is made of styrofoam and is transparent over a large radio frequency range, so that the millimetre wave can penetrate it and detect the corner reflector behind it. In Figure 5b, Domhof et al. [65] designed a styrofoam board with four circular holes and placed a corner reflector at the back. These circular holes are more easily detected by the sparse LiDAR beam since they have no horizontal lines.

The extrinsic calibration of a 4D radar and other sensors can be easily performed by modifying the classical LiDAR to camera calibration methods [66,67]. However, the calibration of conventional radar is a very difficult task, since it returns a 2D point cloud with no elevation resolution. This leads to the problem of vertical misalignment [64], which is defined as the angular deviation between the radar plane and ground plane. Sugimoto et al. [68] moved a corner reflector up and down to the cross-radar plane multiple times. Then, the plane was determined by connecting the peaks with the highest intensity in the sequence. Peršić et al. [64] proposed a two-step optimisation method to mitigate the uncertainty caused by the missing elevation angle. They modelled radar detections as arcs by extending their elevation angle. Similarly, they also converted 3D detections from other sensors to arcs by neglecting the elevation angle. In the first step, they optimised the reprojection error, which is the Euclidean distance of these projected arcs on the ground plane. In the second step, the parameters related to the elevation measurement are refined according to the RCS error. A second-order RCS model was built by fitting RCS measurements with elevation angles. Then, the L2 distance between the expected and measured RCS is minimised. Experiments showed their method enables smaller vertical misalignment than Sugimoto’s method. In order to improve the efficiency, some targetless online calibration approaches [69,70] are proposed to leverage target trajectories for extrinsic estimation.

### 3.3. Data Labelling

Before introducing the labelling process, we first discuss the time synchronisation problem. Different sensors can be synchronised using pulse-per-second (PPS) triggering signals from the GNSS receiver [71]. However, in most of the radar datasets, sensors differ in their triggering time and sampling frequency. Some of them select one sensor as the lead and choose the closest frames from other modalities for synchronisation. Assuming a tolerable time offset of 50 ms, a vehicle with a relative speed of 20 m/s will lead to an offset of 1 m. Therefore, it is necessary to compensate for synchronisation errors in high-speed scenarios. Kaul et al. [72] designed a pose chain method to interpolate inter-frame measurements. The translational and rotational transformations were determined by a constant velocity model and spherical linear interpolation (SLERP) [73], respectively.

Labelling radar data is a difficult task. Both radar point clouds and pre-CFAR data are hard to interpret by human labellers. To reduce the labelling efforts, most of the datasets adopt a semi-automatic labelling framework, which includes two steps: cross-modality pre-labelling and fine-tuning.

In the first step, a well-trained detector on other modalities is leveraged for radar labelling. For 3D tasks, radar point clouds can be annotated by 3D boxes predicted by the detector trained with images and LiDAR point clouds [2]. If we want to obtain pointwise annotations for the radar point cloud, we can first predict a dense semantic map for the corresponding image using a visual segmentation network, such as mask R-CNN [74] or DeepLab V3 [75]. To avoid scale ambiguity, it is better to project the masked image to radar frames using depth measured by LiDAR [72] or a stereo camera [46,47]. Then, each radar point can be associated with the corresponding semantic labels. Pointwise annotation of RAD tensors or RA maps is a similar process. We can firstly use CFAR to detect peaks as detections, then annotate these point detections with the aligned visual semantic map. The CRUW dataset [47] proposes a postprocessing method to obtain pointwise object annotations for RA maps. The authors define an object location similarity (OLS) metric, which jointly considers the similarities in distance, scale, and class. Then, they proposed a location-based non-maximum suppression (NMS) method that selects one object point out of the adjacent points based on the OLS metric. Compared to the RA map, the RD map alone is much more difficult to label. It needs both depth and radial velocity to associate an RD cell with a pixel or a LiDAR point. Radial velocity can be estimated by visual scene flow [76] or by tracking [45].

In the second step, manual inspection is required to correct pre-labelling errors. Identifying radar errors involves domain knowledge and, therefore, requires hiring of radar experts. As a result, building a high-quality, large-scale radar dataset is both time-consuming and financially expensive. To improve labelling efficiency, one way is to reduce the amount of data to be labelled. Dimitrievski et al. [77] leveraged a tracking algorithm to interpolate annotations between key frames. The intermediate position is estimated by a Kalman filter with optical flow as measurements. Meyer et al. [2] adopted an active learning [78] framework to reduce labelling efforts in building the Astyx dataset. The core idea is to only label the most informative data. Specifically, they first labelled a small number of frames and trained a detector with this data subset. The trained detector is then used to make predictions on the remaining unlabelled data. Next, the top N uncertain data are again manually labelled and added to the training subset. This process is repeated until convergence of the validation performance.

### 3.4. Data Augmentation

Data augmentation plays an essential role in improving the generalisation of deep learning models. It is well studied for images [79], LiDAR point clouds [80], and audio spectrograms [81], but overlooked in radar perception. According to the summary report of the Radar Object Detection 2021 (ROD2021) Challenge [82], data augmentation techniques significantly improve the performance of RA-map-based radar detection. Considering the radar representation, we can divide the augmentation techniques into spectral- and point-cloud-based. Augmentation methods can also be featured as local or global depending on whether the entity being augmented is a single object or the entire scene.

Spectral augmentation techniques are used for radar pre-CFAR data. DANet [83] adopts several global augmentation techniques borrowed from computer vision for radar RA maps. The methods include mirroring, resizing, random combination, adding Gaussian noise, and temporal reversing. Although physical fidelity is not explicitly considered, the performance gain proves the effectiveness of these augmentation methods. RADIO [84] implements four types of spectral augmentations, including attenuation, resolution change, adding speckle noise, and background shift. The first two methods are applied to a local patch around each detected object. The attenuation effect is approximated by dampening the cells according to an empirical relationship between the received power and range. The resolution change is modelled by nearest-neighbour interpolation according to the object size. The speckle noise can be approximated as a multiplicative truncated exponential distribution [85] or multiplicative Gaussian noise [84]. Background shift is performed by adding or subtracting a constant value to background cells. RAMP-CNN [86] applies global geometric augmentations to RA maps. It translates and rotates RA maps in Cartesian coordinate, then projects back to the original polar coordinate. The out-of-boundaries areas are cropped off, and the blank areas are filled with background noises. Energy loss and antenna gain loss due to the transformation are compensated according to the radar equation.

Point cloud augmentation aims to introduce invariance to geometric transformations and improve the signal-to-clutter ratio. Compared with spectral augmentation, point cloud augmentation methods can be easily extended to multiple modalities by properly handling occlusion issues [87,88]. Geometric augmentation can be applied locally or globally, depending on whether the transformation is applied to a single target or the entire scene. For radar point clouds, the Doppler velocity and RCS need further consideration. As illustrated in Figure 6, rotating objects locally will affect the radial velocity, and rotating the radar point cloud globally will affect the ego-motion-compensated radial velocity if the ego-motion is not rotated accordingly. Therefore, Palffy et al. [50] advise only using mirroring and scaling along the longitudinal axis as augmentation. Another applicable technique is the copy–paste augmentation, which copies the detected object from other frames and pastes it into the same location in the current frame, as done in [89]. A limitation of these two methods is that they do not change the distribution of detections, while radar points are actually randomly distributed over the object in different frames. According to experiments [90], most of the radar detections are located in the proximity of the vehicle contour and wheel rims. The number of detections per object is inversely proportional to the distance, and the probability of detection on the contour depends heavily on the orientation. Simulation-based methods, which will be introduced in the next section, are more suitable to capture such randomness.

To handle the sparsity issue, many works utilise augmentation to increase the point cloud density. One simple method is accumulating radar points from multiple frames into the current frame. However, accumulation without motion compensation will lead to point cloud aliasing. Long et al. [91] compensated the accumulated radar point cloud with the estimated full velocity, achieving better performance in bounding box regression. Plaffy et al. [50] augmented the accumulated radar point cloud by appending a temporal index to each point as an additional channel. Along with the increased density, this index augmentation is expected to effectively retain the temporal information. Alternatively, Bansal et al. [37] leveraged the space coherence of two radar sensors to increase the point cloud density. They fused point clouds from two radars with overlapping FoVs in a probabilistic manner. They firstly clustered the raw point clouds and then associated clusters from two radars by defining a distance-dependent potential function. Points with low confidence were filtered out as outliers, and the remaining points within the same cluster were coherently accumulated.

### 3.5. Synthetic Data

Synthetic datasets are widely used in computer vision [92,93] and LiDAR perception [94,95] for autonomous driving. Experiments [96] show the networks trained with synthetic data can generalise well in the real-world. By using synthetic radar data, the labelling cost can be completely avoided. Moreover, simulation can be used to generate the safety-critical long-tail scenarios [97]. Physics-based simulation methods, such as ray tracing [98,99], are widely applied to generate synthetic radar point clouds. Experiments [99] show that ray tracing can successfully model the multi-path propagation and separability issue of close objects. However, it is difficult to capture the RCS variation in azimuth with current methods. Another type of simulation is to build a probabilistic model of radar detections, also known as model-based augmentation. The spatial distribution of radar detections over the vehicle can be approximated by the surface–volume model, including the volcanormal measurement model [100], variational Gaussian mixture model (GMM) [100], and hierarchical truncated Gaussian (HTG) [101]. Model parameters can be learned from data. Using this model, we can augment new synthetic radar detections to real point clouds. It is arguable that what level of fidelity is necessary for downstream tasks. In [102], model-based and ray-tracing methods are compared with respect to multiple target tracking. Experiments indicate that the ray-tracing-based model achieves the lowest simulation-to-reality gap.

There are some seminal works utilising learning-based generative models for radar simulation. For example, the deep stochastic radar model [103] adopts a conditional-VAE architecture. The encoder consists of two heads, one for the RAD tensor and one for the object list. The extracted features are concatenated and further processed with an MLP. The decoder generates a radar intensity map in the polar grid conditioned on the encoded feature and random noise. Generative models can also be used in cross-modality data generation, for example GAN-based LiDAR-to-radar generation [104], GAN-based radar-to-image generation [105], and VAE-based radar-to-image generation [106].

## 4. Radar Depth and Velocity Estimation

Radar can measure range and Doppler velocity, but both of them cannot be directly used for downstream tasks. The range measurements are sparse and therefore difficult to associate with their visual correspondences. The Doppler velocity is measured in the radial axis and, therefore, cannot be directly used for tracking. In this section, we summarise depth completion and velocity estimation methods using radar point clouds.

### 4.1. Depth Estimation

Recently, pseudo-LiDAR-based visual object detection [107,108,109] has became a popular research topic. The core idea is to project pixels into a pseudo point cloud to avoid distortions induced by inverse projective mapping (IPM). The pseudo LiDAR detection is built on depth estimation. Visual depth estimation is an ill-posed problem because of the scale ambiguity. However, learning-based methods, either supervised [110] or self-supervised [111], can successfully predict dense depth maps with cameras only. Roughly speaking, these methods learn a priori knowledge of the object size from the data and are therefore vulnerable to some data-related problems, such as sensitivity to input image quality [111] and learning non-causal correlations, such as object and shadow correlations [112]. These limitations can be mitigated with the help of range sensors, such as LiDAR and radar. Depth completion is a sub-problem of depth estimation. It aims to recover a dense depth map for the image using the sparse depth measured by range sensors. Compared to LiDAR, radar has the advantages of a low price, long range, and robustness to adverse weather. Meanwhile, it faces the problems of noisy detections, no height measurements, and sparsity. As shown in Figure 7, due to multi-path propagation, radar can see the non-line-of-sight highly reflective objects, such as wheel rims and occluded vehicles. In [113], the authors refer to this phenomenon as the see0through effect. It is beneficial in 3D coordinates, but brings difficulty in associating radar detections with visual objects in image view.

The two-stage architecture is widely applied for image-guided radar depth completion tasks. Lin et al. [114] adopted a two-stage coarse-to-fine architecture with LiDAR supervision. In the first stage, a coarse radar depth is estimated by an encoder–decoder network. Radar and images are processed independently by two encoders and fused together at the feature level. Then, the decoder outputs a coarse dense depth map in image view. The predicted depth with large errors is filtered out according to a range-dependent threshold. Next, the original sensor inputs and the filtered depth map are sent to a second encoder–decoder to output a fine-grained dense map. In the first stage, the quality of association can be improved by expanding radar detections to better match visual objects. As shown in Figure 8b, Lo et al. [115] applied height extension to radar detections to compensate for the missed height information. A fixed height is assumed for each detection and is projected onto the image view according to the range. Then, the extended detections are sent to a two-stage architecture to output a denoised radar depth map. Long et al. [116] proposed a probabilistic association method to model the uncertainties of radar detections. As shown in Figure 8c, radar points are transformed into a multi-channel enhanced radar (MER) image, with each channel representing the expanded radar depth at a specific confidence level of association. In this way, the occluded detections and imprecise detections at the boundary are preserved, but with a low confidence. Gasperini et al. [113] used radar as supervision to train a monocular depth estimation model. Therefore, they applied a strict filtering to only retain detections with high confidence. In the preprocessing, they removed clutters inside the bounding box that exceeded the range threshold and discarded points in the upper 50% and outer 20% of the box, as well as the overlapping regions to avoid the see-through effect. All the background detections were also discarded. For association, they first applied a bilateral filtering, i.e., an edge-preserving filtering, to constrain the expansion to be within the object boundary. They further clipped the association map close to the edge to get rid of imprecise boundary estimations. To compensate for height information, they directly used the height of the bounding box as a reference. Considering the complexity of the vehicle shape, they extended the detections to the lower third of its bounding box to capture the flat front surface of the vehicle.

As the ground truth, LiDAR has some inherent defects, such as sparsity, limited range, and holes with no reflections. Long et al. [116] suggest to preprocess LiDAR points for better supervision. They accumulated multiple frames of LiDAR point clouds to improve density. Pixels with no LiDAR reaches are assigned zero values. Since LiDAR and the camera do not share the same FoV, the LiDAR points projected to the image view also have the occlusion problem. Therefore, the occluded points are filtered out by two criteria: one is the difference between visual optical flow and LiDAR scene flow, and the other is the difference between the segmentation mask and bounding boxes. Lee et al. [117] suggest to use both the visual semantic mask and LiDAR as supervision signals. Visual semantic segmentation can detect smaller objects at a distance, thus compensating for the limited range of LiDAR. To extract better representations, they leveraged a shared decoder to learn depth estimation and semantic segmentation concurrently. Both the LiDAR measurement and the visual semantic mask annotations are used as supervision. Accordingly, the loss function consists of three parts: a depth loss with LiDAR points as the ground truth, a visual semantic segmentation loss, and a semantic guided regularisation term for smoothness.

Projecting radar to the image view will lose the advantages of the see-through effect. Alternatively, Niesen et al. [118] leveraged radar RA maps for depth prediction. They used a short-range radar with a maximum range of 40 m. Because of the low angular resolution, the azimuth smearing effect is obvious, i.e., the detections are smeared as a blurry horizontal line in RA maps. It is expected that fusion of the image and RA map can mitigate this effect. Therefore, they used a two-branch encoder–decoder network with the radar RA map and image as inputs. A dense LiDAR depth map was used as the ground truth. Different from the above methods that align LiDAR to the image, they cropped, downsampled, and quantised LiDAR detections to match the radar’s FoV and resolution. The proposed method was tested with their self-collected data. Although the effectiveness of the RA map and point cloud was not compared, it provides a new direction to explore radar in the depth estimation task.

### 4.2. Velocity Estimation

For autonomous driving, velocity estimation is helpful for trajectory prediction and path planning. Radar can accurately measure the Doppler velocity, i.e., radial velocity in polar coordinates. If a vehicle moves parallel to the ego-vehicle at a distance, its actual velocity can be approximated by the measured Doppler velocity. However, this only applies in highway scenarios. On urban roads, it is possible for an object to move tangentially while crossing the road, then its Doppler velocity will be close to zero. Therefore, Doppler velocity cannot replace full velocity. Recovering full velocity from the Doppler velocity needs two steps: first, compensate the ego-motion, then estimate the tangential velocity. In the first step, the ego-motion can be estimated by visual-inertial odometry (VIO) and GPS. Radar-inertial odometry [119,120] can also be used in visually degraded or GPS-denied environments. Then, the Doppler velocity is compensated by subtracting the ego-velocity. In the second step, the full velocity is estimated according to the geometric constraints. Suppose that the radar observes several detections of an object and that the object is in linear motion. As shown in Figure 9a, the relationship between the predicted linear velocity $\left(v_{x},v_{y}\right)$ and the measured Doppler velocity $v_{r,i}$ is given by
(7)$$v_{r,i}=v_{x}\cos\left(\theta_{i}\right)+v_{y}\sin\left(\theta_{i}\right)$$
where the subscript *i* denotes the *i*-th detection and $\theta_{i}$ is the measured azimuth angle. By observing *N* detections per object, we can solve the linear velocity using the least-squares method. However, the L2 loss is not robust to outliers, such as clutter and the mirco-Doppler motion of wheels. Kellner et al. [121] applied RANSAC to remove outliers, then used orthogonal distance regression (ODR) to find the optimal velocity.

Although the linear motion model is widely used for its simplicity, it will generate large position errors for motion with high curvature [122]. Alternatively, as shown in Figure 9b, the curvilinear motion model is given by
(8)$$v_{r,i}=\omega \left(y_{c}-y^{S}\right)\cos\left(\theta_{i}\right)-\omega \left(x_{c}-x^{S}\right)\sin\left(\theta_{i}\right)$$
where ω is the angular velocity, θ is the angle of the detected point, $\left(x_{c},y_{c}\right)$ represents the position of the instantaneous centre of rotation (ICR), and $\left(x^{S},y^{S}\right)$ represents the known radar position. In order to decouple angular velocity and the position of the ICR, we need at least two radar sensors that observe the same object. Then, we can transform (Equation 8) into a linear form as
(9)$$y_{j}^{S}\cos\left(\theta_{ji}\right)-x_{j}^{S}\sin\left(\theta_{ji}\right)=y_{c}\cos\left(\theta_{ji}\right)-x_{c}\sin\left(\theta_{ji}\right)-v_{ji}^{D}\omega^{-1}$$
where the subscript *j* denotes the *j*-th radar. Similarly, RANSAC and ODR can be used to find the unbiased solution of both the angular velocity and position of the ICR [123]. For the single radar setting, it is also possible to derive a unique solution of (Equation 8) if we can correctly estimate the vehicle shape. According to the Ackermann steering geometry, the position of the ICR should be located on a line extending from the rear axle. By adding this constraint to (Equation 8), the full velocity can be determined in closed form [124].

The above methods predict velocity at the object level under the assumption of rigid motion. However, the micro-motion of object parts, such as the swinging arms of pedestrians, are also useful for classification. Capturing these non-rigid motions requires velocity estimation at the point level. This can be achieved by fusing with other modalities or by using temporal consistency between adjacent radar frames. Long et al. [91] estimated pointwise velocity by the fusion of radar and cameras. They first estimated the dense global optical flow and the association between radar points and image pixels through neural network models. Next, they derived the closed-form full velocity based on the geometric relationship between optical flow and Doppler velocity. Ding et al. [125] estimated the scene flow for the 4D radar point cloud in a self-supervised learning framework. Scene flow is a 3D motion field and can be roughly considered as the linear velocity field. Their model consists of two steps: flow estimation and static flow refinement. In the flow estimation step, they adopted a similar structure with PointPWCNet [126]. To compensate for the positional randomness of detections between frames, a cost–volume layer is utilised for patch-to-patch correlation. The features and correlation maps are then sent to a decoder network for flow regression. In the static flow refinement step, they assumed that most radar detections are static and, therefore, used the Kabsch algorithm [127] to robustly estimate the ego-motion. They then filtered out moving objects based on the coarse ego-motion and applied the Kabsch algorithm again to all static points for fine-grained ego-motion estimation. The self-supervised loss consists of three parts: a radial displacement loss, which penalises errors between the estimated velocity projected along the radial axis and the measured Doppler velocity, a soft Chamfer distance loss, which encourages temporal consistency between two consecutive point clouds, and a soft spatial smoothness loss, which encourages the spatial consistency for the estimated velocities with their neighbours. The soft version of loss is used to model spatial sparsity and the temporal randomness of the radar point cloud.

## 5. Radar Object Detection

Due to low resolution, the classical radar detection algorithm has limited classification capability. In recent years, the performance of automotive radar has greatly improved. At the hardware level, next-generation imaging radars can output high-resolution point clouds. At the algorithmic level, neural networks show their potentials to learn better features from the dataset. In this section, we consider a broader definition of radar detection, including pointwise detection, 2D/3D bounding box detection, and instance segmentation. We first introduce the classical detection pipeline and recent improvements on clustering and feature selection. As shown in Figure 10, neural networks can be applied to different stages in the classical pipeline. According to the input data structure, we classify the deep radar detection into point-cloud-based and pre-CFAR-based. Radar point cloud and pre-CFAR data are similar to the LiDAR point cloud and visual image, respectively. Accordingly, the architectures for LiDAR and vision tasks can be adapted for radar detection. We focus on how knowledge from the radar domain can be incorporated into these networks to address the low SNR problem.

### 5.1. Classical Detection Pipeline

As shown in Figure 10, the conventional radar detection pipeline consists of four steps: CFAR detection, clustering, feature extraction, and classification. Firstly, a CFAR detector is applied to detect peaks in the RD heat map as a list of targets. Then, the moving targets are projected to Cartesian coordinates and clustered by DBSCAN [19]. Static targets are usually filtered out before clustering because they are indistinguishable from environmental clutter. Within each cluster, hand-crafted features, such as the statistics of measurements and shape descriptors, are extracted and sent to a machine learning classifier. Improvements can be made upon each of these four steps. CFAR is usually executed in an on-chip DSP, so the choice of method is restricted by hardware support. Cell-averaging (CA) CFAR [18] is widely used due to its efficiency. It estimates the noise as the average power of neighbouring cells around the cell under test (CUT) within a CFAR window. A threshold is set to achieve a constant false alarm rate for Rayleigh-distributed noise. The next-generation high-resolution radar chips also support order-statistics (OS) CFAR [18]. It sorts neighbouring cells around the CUT according to the received power and selects the k-th cell to represent the noise value. OS-CFAR has advantages in distinguishing close targets, but introduces a slightly increased false alarm rate and additional computational costs. More sophisticated CFAR variants are summarised in [128], but are rarely used in automotive applications. Deep learning methods can be used to improve noise estimation [129] and peak classification [128] in CFAR. Clustering is the most important stage in the radar detection pipeline, especially for the next-generation high-resolution radar [130]. DBSCAN is favoured for several reasons: it does not require a pre-specified number of clusters; it fits arbitrary shapes; it runs fast [131]. Some works improved DBSCAN by explicitly considering the characteristics of radar point clouds. Grid-based DBSCAN [132] suggests clustering radar points in an RA grid map to avoid the range-dependent resolution variations in Cartesian coordinates. Multi-stage clustering [133] proposes a coarse-to-fine two-stage framework to alleviate the negative impact of clutter. It applies a second cluster merging based on the velocity and spatial trajectory of clusters estimated from the first stage.

With the improvement of automotive radar resolution, radar target classification has become a hot research topic. For moving objects, the micro-Doppler velocity of moving components such as wheels and arms can be useful for classification. To better observe these micro-motions, short-time Fourier transform (STFT) is applied to extract Doppler spectrograms. Different types of VRUs can be classified according to their micro-Doppler signatures [134,135]. For static objects, Cai et al. [136] suggest the use of statistical RCS and time-domain RCS as useful features for classification of vehicles and pedestrians. Some researchers work on exploiting a large number of features for better classification. Scheiner et al. [137] considered a large set of 98 features and used the heuristic-guided backward elimination for feature selection. They found that range and Doppler features are most important for classification, while angle and shape features are usually discarded, probably because of the low angular resolution. Schumann et al. [138] compared the performance of random forest and LSTM for radar classification. Experiments showed that LSTM with an input of eight-frame sequences performs slightly better than random forests, especially in the classification of classes with a similar shape, such as pedestrians and pedestrian groups, and for false alarms. However, LSTM is more sensitive to the amount of training examples. To cope with class imbalance in radar datasets, Scheiner et al. [139] suggest using classifier binarisation techniques, which can be divided into two variants: one-vs.-all (OVA) and one-vs.-one (OVO). OVA trains *N* classifiers to separate one class from the other N−1 classes, and OVO trains N2 classifiers for every class pair. During inference, the results are decided by max-voting.

### 5.2. Point Cloud Detector

End-to-end object detectors are expected to replace the conventional pipelines based on hand-crafted features. However, the convolutional neural network is not well designed for sparse data structure [140]. It is necessary to increase the input density of the radar point cloud for better performance. Dreher et al. [141] accumulated radar points into an occupancy grid mapping (OGM), then applied YOLOv3 [142] for object detection. Some works [143,144,145] utilise point cloud segmentation networks, such as PointNet [146] and PointNet++ [147], followed by a bounding box regression module for 2D radar detection. The original 3D point cloud input is replaced by a 4D radar point cloud with two spatial coordinates in the x-y plane, Doppler velocity, and RCS. Scheiner et al. [145] compared the performances of the two-stage clustering method, OGM-based method, and PointNet-based method with respect to 2D detection. Experiments showed that the OGM-based method performs best, while the PointNet-based method performs far worse than others probably due to sparsity. Liu et al. [148] suggest that incorporating global information can help with the sparsity issue of the radar point cloud. Therefore, they added a gMLP [149] block to each set abstraction layer in PointNet++. The gMLP block is expected to extract global features at an affordable computational cost.

Most radar detection methods only apply to moving targets, since static objects are difficult to classify due to low angular resolution. Schumann et al. [150] propose a scene understanding framework to detect both static and dynamic objects simultaneously. For static objects, they first built an RCS histogram grid map through the temporal integration of multiple frames and send it to a fully convolutional network (FCN) [151] for semantics segmentation. For dynamic objects, they adopted a two-branch recurrent architecture: One is the point feature generation module, which uses PointNet++ to extract features from the input point cloud. The other is the memory abstraction module, which learns temporal features from the temporal neighbours in the memorised point cloud. The resulting features are concatenated together and sent to an instance segmentation head. In addition, a memory update module is proposed to integrate targets into the memorised point cloud. Finally, static and dynamic points are combined into a single semantic point cloud. The proposed framework can successfully detect moving targets such as cars and pedestrians, as well as static targets such as parked cars, infrastructures, poles, and vegetation.

As 4D radars have gradually come to the market, radar point cloud density has increased considerably. A major advantage of 4D radar is that static objects can be classified based on elevation measurements without the need to build an occupancy grid map. Therefore, it is possible to train a single detector for both static and dynamic objects. Plaffy et al. [50] applied PointPillars [152] to 4D radar point clouds for 3D detection of multi-class road users. They found the performance can be improved by temporal integration and by introducing additional features, such as elevation, Doppler velocity, and RCS. Among them, the Doppler velocity is essential for detecting pedestrians and bicyclists. However, the performance of the proposed 4D radar detector (mAP 47.0) is still far inferior to their LiDAR detector on 64-beam LiDAR (mAP 62.1). They argue this performance gap comes from radar’s poor ability in determining the exact 3D position of objects. RPFA-Net [153] improves PointPillars by introducing a radar pillar features attention (PFA) module. It leverages self-attention to extract the global context feature from pillars. The global features are then residually connected to the original feature map and sent to a CNN-based detection network. The idea behind this is to explore the global relationship between objects for a better heading angle estimation. In fact, self-attention is basically a set operator, so it is well suited for sparse point clouds. Radar transformer et al. [154] is a classification network constructed entirely of self-attention modules. The 4D radar point cloud is first sent to an MLP network for input embedding. The following feature extraction network consists of two branches. In the local feature branch, it uses three stacked set abstraction modules [147] and vector attention modules [155] to extract hierarchical local features. In the global feature branch, the extracted local features at each hierarchy are concatenated with the global feature map at the previous hierarchy and fed into a vector attention module for feature extraction. In the last hierarchy, a scalar-attention, i.e., the conventional self-attention, is used for feature integration. Finally, the feature map is sent to a classification head. Experiments showed the proposed radar transformer outperforms other point cloud networks in terms of classification. The above two attention-based approaches show their potential in modelling the global context and extracting semantic information. Further works should focus on combining these two advantages into a fully attention-based detection network.

### 5.3. Pre-CFAR Detector

There are some attempts to explore the potential of pre-CFAR data for detection. Radar pre-CFAR data encode rich information of both targets and backgrounds, but this is hard to interpret by humans. Neural networks are expected to better utilise this information. One option is to use neural networks to replace CFAR [156] or DOA estimation [76,157]. Readers can refer to [158] for a detailed survey of learning-based DOA estimation. Alternatively, there are also some efforts to perform end-to-end detection through neural networks. The deep radar detector [159] jointly trains two cascaded networks for CFAR and DOA estimation, respectively. Zhang et al. [160] used stacked complex RD maps as the input to an FCN for 3D detection. In order to remove the DC component in phase, they performed a phase normalisation by using RD cells in the first receiver as normalisers. They argued that phase normalisation is crucial for successful training. Rebut et al. [51] designed a DDM-MIMO encoder with a complex RD map as the input. In the DDM configuration, as illustrated in Figure 3, all Tx antennas transmit signals at the same time. Instead of performing waveform separation, they directly applied range FFT and Doppler FFT to ADC signals received by Rx antennas. In this way, targets detected from different Tx antennas should be located separately with fixed Doppler shifts in the RA map. To extract these features, they designed a two-layer MIMO encoder, consisting of a dilated convolutional layer to separate Tx channels, followed by a convolutional layer to mix the information. This MIMO encoder was jointly trained with the following RA encoder, detection head, and segmentation head.

In close-field applications that require large bandwidth and high resolution, RD maps are not suitable because the extended Doppler profile can lead to false alarms. The RA map, on the other hand, does not suffer from the same problem. For each detection point on the RA map, the micro-Doppler information in slow time can be utilised for better classification. RODNet [47] uses complex RA maps as input for object detection. It performs range FFT followed by angle FFT to obtain a complex RA map for each sampled chirp. It is difficult to separate static clutter and moving objects using the RA map alone without Doppler dimension. To utilise the motion information, it samples a few chirps within a frame. Then, the sequences of RA maps corresponding to these chirps are sent to a temporal convolution layer. Specifically, it first uses 1 × 1 convolutions along the chirp dimension to aggregate temporal information. Then, a 3D convolution layer is used to extract temporal features. Finally, the features are merged along the chirp dimension by max-pooling. Experiments indicate sampling 8 chirps out of 255 can achieve a comparable performance with using the full chirp sequences.

Training neural network to utilise phase information in complex RA or RD maps is a difficult task. Alternatively, some works attempt to use the real-valued RAD tensor as the input. A key issue in using the 3D RAD tensor as the input is the curse of dimensionality. Therefore, many techniques are proposed to reduce the computational cost of 3D tensor processing. RADDet [46] normalises and reshapes the RAD tensor to an image-like data structure. The Doppler dimension is treated as the channel of 2D RA maps. Then, YOLO is applied to the RA map for object detection. One disadvantage is that this method fails to utilise the spatial distribution of Doppler velocities. Alternatively, 3D convolution can be used to extract features from all three dimensions in a 3D tensor, but requires huge computation and memory overheads [161]. RODNet [47] samples chirp sequences, as described above, to reduce input dimensionality. RTCNet [162] reduces tensor size by cropping a small cube around each point detected by CFAR and then uses 3D CNN to classify these small cubes. However, its detection performance is limited by the CFAR detector. To fully exploit the information encoded in RAD tensors, some works [86,163,164] adopt the multi-view encoder–decoder architecture. Major et al. [163] and Ouaknine et al. [164] both utilised a similar multi-view structure. The RAD tensor is projected into three 2D views. Then, three decoders extract features from these views, respectively. To fuse these features, Ouaknine et al.directly concatenated three feature maps. Major et al.recovered the tensor shape by duplicating these 2D feature maps along the missing dimension, then used a 3D convolution layer to fuse them. Next, the Doppler dimension is suppressed by pooling to recover the shape of the RA feature map. Finally, the fused feature maps are sent to a decoder for downstream segmentation tasks. Another difference is Major et al.used a skip-connection, while Ouaknine et al.adopted an ASPP [75] pathway to encode information from different resolutions. RAMP-CNN et al. [86] is also built in a multi-view architecture, but it uses three encoder–decoders for feature map extraction. Their fusion method is similar to Major’s, but in 2D.

Radar pre-CFAR data are captured in polar coordinates. For object detection, polar-to-Cartesian transformation is necessary to obtain the correct bounding box. Major et al. [163] compared three configurations for coordinate transformation: preprocessed input transformation, learning from neural networks, and transformation on a middle-layer feature map. Experiments showed applying explicit polar-to-Cartesian transformation to the last-layer feature map achieves the best performance, the implicit learning-based transformation is slightly worse, and the preprocessed transformation is far inferior to the other two. They attributed this poor performance to distorted azimuth sidelobes in the input. In fact, conventional 2D convolution is not the best choice for radar pre-CFAR data, since the range, Doppler, and azimuth dimension vary in their dynamic ranges and resolutions. Instead of 2D convolution, PolarNet [165] uses a cascade of two 1D convolutions, including a columnwise convolution to extract range-dependent features, followed by a row-wise convolution to mix information from spatial neighbours. A similar idea is used in Google’s RadarNet [166] for gesture recognition. They first extracted rangewise features, then summarised them together in the later stage. Meyer et al. [167] used an isotropic graph convolution network (GCN) [168] to encode the RAD tensor and achieved more than a 10% improvement in AP for 3D detection. They argued that the performance gain comes from the ability of GCN to aggregate information from neighbouring nodes.

Incorporating temporal information is an effective way to improve the performance of pre-CFAR detectors. There are multiple ways to add temporal information to the network. Major et al. [163] used a convolutional LSTM layer to process a sequence of feature maps from the encoder network. Experiments indicated the temporal layer enables more accurate detection and significantly better velocity estimation. Ouaknine et al. [164] compared the performance between the static model with accumulated inputs and the temporal model with stacked inputs. For the static model, RAD tensors within three frames are accumulated into one single tensor and fed to a multi-view encoder–decoder for segmentation. For the temporal model, RAD tensors within five frames are stacked to form a 4D tensor and then sent to a multi-view encoder–decoder. In each branch, multiple 3D convolution layers are used to leverage spatial–temporal information. The results show that the introduction of the temporal dimension can significantly improve detection performance. Pervsic et al. [69] discussed the effect of the number of stacked radar frames. They found too long frames will introduce many background clutter, which in turn makes it difficult for the model to learn target correspondences. According to their experiments, stacking of five frames is the most suitable choice. RODNet [47] investigates stacking multiple frames at the feature level. It concatenates the extracted per-frame features and sends them to a 3D CNN layer. For motion compensation, they applied deformable convolution [169] on the chirp dimension in the first few layers. In addition, an inception module with different temporal lengths was used in the later layers. Despite the introduction of additional computational costs, these two temporal modules significantly improve the average precision. Li et al. [170] explicitly modelled the temporal relationship between features extracted from two consecutive frames using an attention module. Firstly, they stacked RA maps in two orders, i.e., current frame on top and previous frame on top. Then, they used two encoders to extract features from these two inputs and concatenated the features together. A positional encoding was further added to compensate the positional imprecision. Next, the features were sent to a masked attention module. The mask was used to disable cross-object attention in the same frame. Finally, the temporally enhanced features were sent to an encoder for object detection. This attention-based approach is more semantically interpretable and avoids the locality constraint induced by convolution.

## 6. Sensor Fusion for Detection

Different sensors observe and represent an object with different features. Sensor fusion can be considered as the mapping of different modalities into a common latent space where different features of the same object can be associated together. In this section, we focus on sensor fusion for detection. We argue that the conventional taxonomy of fusion architectures into early (input), middle (feature), and late (decision) fusion is ambiguous for neural-network-based detection. For example, in the definition of late fusion, we cannot distinguish between ROI-level (without category information) fusion and object-level (with category information) fusion. Therefore, we explicitly classify fusion methods according to the fusion stage. This is beneficial because different fusion stages correspond to different levels of semantics, i.e., the classification capabilities. As shown in Figure 11, we classify fusion architectures into four categories: input fusion, ROI fusion, feature map fusion, and decision fusion.

### 6.1. Input Fusion

Input fusion is applied to the radar point cloud. It projects radar points into a pseudo-image with the range, velocity, and RCS as channels [171,172]. Then, similar to an RGB-depth image, the radar pseudo-image and the visual image are concatenated as a whole. Finally, a visual detector can be applied to this multi-channel image for detection. Input fusion does not make independent use of the detection capability of radar. In other words, the radar and vision modalities are tightly coupled. Assuming good alignment between modalities, it makes it easier for the network to learn joint feature embeddings. However, an obvious disadvantage is that the architecture is not robust to sensor failures.

The fusion performance depends on the alignment of radar detections with visual pixels. As mentioned in Section 4.1, the difficulties lie in three aspects: Firstly, the radar point cloud is highly sparse. Many reflections from the surface are bounced away due to specular reflections. As a result, the detected points are sparsely distributed over the object. In addition to the sparsity, the lateral imprecision of radar measurements leads to further difficulties. The radar points can be out of the visual bounding box. The imprecision comes from different aspects, e.g., imprecise extrinsic calibration, multi-path effects, and low angular resolution. The third limitation is that low-resolution radar does not provide height information. To address these difficulties, some association techniques are required. Relying on the network to implicitly learn association is a hard task, because the network tends to simply ignore the weak modality, such as radar. The expansion methods described in Section 4.1 can be applied as a preprocessing stage for input fusion. However, object detection does not require such a strict association as depth completion, so some of the expansion methods are too costly for real-time processing. Nobis et al. [171] utilised the lightweight height extension as preprocessing. Both Chadwick et al. [172] and Yadav et al. [173] added a one-layer convolution to radar input before concatenation. This convolutional layer can be considered as a lightweight version of the association network. Radar detections at different ranges require different sizes of the receptive field for association. Therefore, Nobis et al. [171] concatenated the radar pseudo-image with image feature maps at multiple scales.

### 6.2. ROI Fusion

ROI fusion is adapted from the classical two-stage detection framework [174]. Regions of interest (ROIs) can be considered as a set of object candidates without category information. The fusion architecture can be further divided into cascade fusion and parallel fusion. In cascade fusion, radar detections are directly used for region proposal. Radar points are projected into image view as the candidate locations for anchors. Then, the ROI is determined with the help of visual semantics. In the second stage, each ROI is classified and its position is refined. Nabati et al. [175] adopted two techniques to improve the anchor quality. They added offsets to anchors to model the positional imprecision of radar detections. To mitigate the scale ambiguity in the image view, they rescaled the anchor size according to the range measurements. In their following work [176], they directly proposed 3D bounding boxes and then mapped these boxes to the image view. In this way, the rescaling step can be avoided. It is also possible to propose the region on the radar point cloud using visual ROIs. For example, CenterFusion [177] proposed a frustum-based association to generate radar ROI frustums using visual bounding boxes.

Cascade fusion is particularly well suited for low-resolution radars, where the radar point cloud has a high detection recall, but is very sparse. However, there are two potential problems with the cascade structure. Firstly, the performance is limited by the completeness of the proposed ROIs in the first stage. In other words, if an object is missed, we cannot recover it in the second stage. The second problem is that the cascade structure cannot take advantage of modality redundancy. If the radar sensor is nonfunctional, the whole sensing system will fail. Therefore, it is necessary to introduce a parallel structure to ROI fusion. Nabati et al. [176] adopted a two-branch structure for ROI fusion. The radar and visual ROIs are generated independently. Then, the fusion module merges radar ROIs and visual ROIs by taking a set union, while the redundant ROIs are removed through NMS. To enable the adaptive fusion of modalities, Kim et al. [178] proposed a gated region of interest fusion (GRIF) module for ROI fusion. It first predicts a weight for each ROI through a convolutional sigmoid layer. Then, the ROIs from radar and vision are multiplied by their corresponding weights and elementwise added together.

### 6.3. Feature Map Fusion

Feature-map fusion leverages the semantics from both radar and images. From Section 5.3, we find that high-resolution radars can provide sufficient semantic cues for classification. Therefore, feature map fusion utilises two encoders to map radar and images into the same latent space with high-level semantics. The detection frameworks are flexible, including one-stage methods [179,180] and two-stage methods [33,181,182]. The one-stage methods leverage two branches of neural networks to extract feature maps from radar and images, respectively, and then concatenate the feature maps together. The two-stage fusion methods are adapted from the classical fusion architecture AVOD [183]. They firstly fuse the ROIs proposed from the radar and image in the first stage. In the second stage, the fused ROIs are projected to the radar and visual feature maps, respectively. The feature maps inside the ROIs are cropped and resized to an equal-sized feature crop. The feature crop pairs from the radar and image are then fused by the elementwise mean and sent to a detection head. Generally speaking, the two-stage method has better performance, but it is much slower than the one-stage method. Anchor-free methods [184,185] further avoid the complicated computation related to anchor boxes, such as calculating the IOU score during training.

Feature map fusion allows the network to flexibly combine radar and visual semantics. However, the fusion network may face the problem of overlooking weak modalities and modality synergies [186]. Some training techniques are needed to force the network to learn from radar input. Nobis et al. [171] adopted a modalitywise dropout approach that randomly deactivates the image branch during training. Lim et al. [179] used a weight freezing strategy to fix the weights of the pre-trained feature extractors when training the fusion branch. Experiments show that freezing only the image branch works best. However, the fusion of multiple modalities is not guaranteed to always be better than using a single modality. Sometimes, we want the network to lower the weight of the radar branch if it gives noisy inputs. To achieve adaptive fusion, Cheng et al. [187] adopted self-attention and global channel attention [188] in their network. The self-attention is used to enhance real target points and weaken clutter points. Then, the global attention module is applied to estimate modalitywise weights. Bijelic et al. [43] estimated the sensor entropy as the modality weight. For each modality, the entropy was evaluated pixelwise as a weight mask. Then, these weight masks are multiplied with the corresponding feature maps at each fusion layer.

### 6.4. Decision Fusion

Decision fusion assumes that objects are detected independently by different modalities and fuses them according to their spatial–temporal relationships. This structure realises sensing redundancy at the system level and is therefore robust to modalitywise error. Due to the low resolution of radar, most existing studies do not explicitly consider the category information estimated by radar. In other words, they only fuse the location information from radar and vision branches, while retaining the category information estimated by vision. Since the next-generation 4D radar can provide classification capabilities, it is expected that future fusion frameworks should consider both location and category information.

The location can be optimally fused in a tracking framework. Different objects are first associated and then sent to a Bayesian tracking module for fusion. Due to the low resolution of radar, association is difficult to achieve in some scenarios, e.g., a truck splitting into two vehicles or two close objects merging into one. Such association ambiguity can be mitigated using a track-to-track fusion architecture [189]. By estimating tracks, temporal information can be leveraged to filter out false alarms and interpolate missed detections. Some researchers exploit deep learning to make a better association between radar and other modalities. RadarNet [184] proposed an attention-based late fusion to optimise the estimated velocity. Firstly, they trained a fiver-layer MLP with softmax to estimate the normalised association scores between each bounding box and its nearby radar detections. Then, they predicted the velocity by weighted averaging of the radar-measured velocities using the association scores. AssociationNet [190] attempts to map the radar detections to a better representation space in the contrastive learning framework. It first projects radar objects and visual bounding boxes to the image plane as pseudo images. To utilise the visual semantics, they concatenated these pseudo images with the original image. Next, the concatenated images are sent to an encoder–decoder network to output a feature map. Representation vectors are extracted from the feature map according to the locations of radar detections. A contrastive loss is designed to pull together the representation vectors of positive samples and push away the representation vectors of negative examples. During inference, they compute the Euclidean distance between the representation vectors of all possible radar–visual pairs. The pairs with a distance below the threshold are considered associative.

Category information, especially the conflict in category predictions, is difficult to handle in sensor fusion. BayesOD [191] proposes a probabilistic framework for fusing bounding boxes with categories. The locations of bounding boxes are modelled by Gaussian distributions. The category prior is modelled as a Dirichlet distribution, thereby allowing a Dirichlet posterior to be computed in closed form. Then, the bounding box with the highest categorical score is considered as the cluster centre, while the other bounding boxes are treated as measurements. Finally, Bayesian inference is used to optimally fuse the location and category information of these bounding boxes. Probabilistic methods have their inherent shortage in modelling the lack of knowledge [192]. For example, a uniform distribution brings confusion if either the network has no confidence in its prediction or the input is indeed ambiguous for classification. In contrast, set-based methods have no such problem. Chavez et al. [193] leveraged the evidential theory to fuse the LiDAR, camera, and radar. They considered the frame of discernment, i.e., the set of mutually exclusive hypotheses, as Ω={pedestrians(p),bikes(b),cars(c),truck(t)}, and assigned each possible hypothesis, i.e., a subset of Ω, with a belief. In the case of object detection, possible hypotheses are selected according to sensor characteristics. For example, a car is sometimes confused as part of a truck. Thus, if a car is detected, evidence should be also put into the set {c,t} and the set of ignorance Ω. Accordingly, we can assign the belief *m* to a car detection as
(10)$$m\left(\left\{c\right\}\right)=\gamma_{c}\alpha_{c},\;m\left(\left\{c,t\right\}\right)=\gamma_{c}\left(1-\alpha_{c}\right),\;m\left(\Omega \right)=1-\gamma_{c}$$
where $\gamma_{c}$ is a discounting factor to model the uncertainty of misdetection and $\alpha_{c}$ is the accurateness, i.e., the rate of correct predictions in car detecting. Suppose there are two sources of evidence $S_{1}$ and $S_{2}$ from different modalities. Each of these sources provides a list of detections as $A=\{a_{1},a_{2},\ldots ,a_{m}\}$ and $B=\{b_{1},b_{2},\ldots ,b_{n}\}$. Then, three propositions can be defined regarding the possible association of two detections $a_{i}$ and $b_{j}$ as:{1} if $a_{i}$ and $b_{j}$ are the same object;{0} if $a_{i}$ and $b_{j}$ are not the same object{0,1} for the ignorance of association.

The belief of association can be determined according to both location and category similarities. The evidence for location similarity is defined according to the Mahalanobis distance as
(11)$$\begin{array}{cc} \; & m_{a_{i},b_{j}}^{p}\left(\left\{0\right\}\right)=\alpha \left(1-f\left(d_{a_{i},b_{j}}\right)\right)m_{a_{i},b_{j}}^{p}\\ \; & m_{a_{i},b_{j}}^{p}\left(\left\{1\right\}\right)=\alpha f\left(d_{a_{i},b_{j}}\right)\;m_{a_{i},b_{j}}^{p}\left(\left\{1,0\right\}\right)=1-\alpha \end{array}$$
where $f\left(d_{a_{i},b_{j}}\right)=\exp\left(-\lambda d_{a_{i},b_{j}}\right)\in \left[0,1\right]$ measure the similarity with respect to the Mahalanobis distance $d_{a_{i},b_{j}}$ and a scaling factor λ and α is an evidence discounting factor. For the category similarity, two detections belonging to the same category is too weak to provide evidence that they are the same object. However, if two detections are of different categories, it is reasonable to assign evidence to the proposition that they are not the same object. Accordingly, the evidence for category similarity is given by
(12)$$\begin{array}{cc} \; & m_{a_{i},b_{j}}^{c}\left(\left\{0\right\}\right)=\sum_{A\cap B=\emptyset }m_{a_{i}}^{c}\left(A\right)m_{b_{j}}^{c}\left(B\right)\;\forall A,B\subset \Omega \\ \; & m_{a_{i},b_{j}}^{c}\left(\left\{1\right\}\right)=0,\;m_{a_{i},b_{j}}\left(\left\{0,1\right\}\right)=1-m_{a_{i},b_{j}}^{c}\left(\left\{0\right\}\right) \end{array}$$
where the mass evidence is fused if no common category hypothesis is shared, i.e., A∩B=∅. The rest of the evidence is placed in the ignorance hypothesis. Finally, for each detection pair, the category similarity and the location similarity are fused according to Yager’s combination rule [194]. Evidential fusion provides a reliable framework for information fusion. However, it cannot be directly applied to neural-network-based detectors that make predictions on a single hypothesis. To address this problem, conformal prediction [195] can be used to generate confidence sets from a trained network using a small amount of calibration data.

## 7. Challenges

Although deep radar perception shows good performance on datasets, there are few studies investigating the generalisation of these methods. In fact, some challenging situations are overlooked, but may prohibit the use of these methods in real-world scenarios. For example, the ghost objects caused by multi-path propagation are common in complex scenarios. Over-confidence is a general problem with neural networks. Since radar is always used for safety-critical applications, it is important to calibrate the detection network and output the predictive uncertainty. Even though we always refer to radar as an all-weather sensor, robustness in adverse weather is not well tested in many radar fusion methods. In this section, we present these three challenges and summarise some recent works that attempt to solve them.

### 7.1. Ghost Object Detection

Multi-path is a phenomenon in the physics of waves where a wave from a target travels to a detector through two or more paths. Because of multi-path propagation, the radar receives both direct reflections and indirect time-shifted reflections of targets. If the target reflections and the multi-path reflections occupy the same RD cell, the performance of DOA estimation is affected. Otherwise, if they occupy different cells, it can produce ghost targets in multi-path directions. In the latter case, since ghost detection has similar dynamics to the real target, it is difficult to eliminate them in the traditional detection pipeline. The multi-path effect can be classified into three types [196]. The first type is the reflection between ego-vehicle and targets. Therefore, the distance and velocity of clutter should be multiple times the true measurement. The second type is the underbody reflection. It usually happens under the truck, resulting in points with longer distances. This see-through effect is sometimes beneficial, since occluded vehicles can be detected. The third type is mirrored ghost detections caused by the reflective surface. Because of the large wavelength of automotive 77 GHz radar, many flat facilities, such as concrete walls, guardrails, and noise cancellation walls, can be regarded as reflective surfaces. As shown in Figure 12, this kind of multi-path effect can be further categorised into type 1 and type 2 depending on whether the final reflection occurs on the target or the surface [197]. The number of reflections is referred to as the order of the multi path. Usually, only orders below 3 need to be considered, since higher-order reflections return little energy due to signal diffusion.

A high-quality dataset is necessary for the performance evaluation of ghost detection. However, the labelling of ghost objects is a difficult task and requires expert knowledge. Chamseddine et al. [89] proposed a method to automatically identify radar ghost objects by comparing with the LiDAR point cloud. However, LiDAR measurements are not perfect. They have their own inherent defects, such as sparsity, limited range, and holes with no reflections. Therefore, using LiDAR as the ground truth could be sometimes problematic. In the Radar Ghost dataset [61], ghost objects are manually annotated with the help of a helper tool. This tool can automatically calculate the locations of potential ghosts based on real objects and reflective surfaces. As a result, four types of multi-path effects are annotated, including type-1 second-order bounces, type-2 second-order bounces, type-2 third-order bounces, and the other higher-order bounces. In addition, they also provide a synthetic dataset by overlaying objects from different frames within the same scene.

Unlike clutter, ghost objects cannot be filtered by temporal tracking because they have the same kinematic properties as real targets. Instead, they can be detected by geometric methods [196,198]. With a radar ghost dataset, it is also possible to train a neural network for ghost detection, such as PointNet-based methods [89] and PointNet++-based methods [197,199]. Because of the signal diffusion, the higher-order reflections can be safely ignored. Thus, ghost objects usually occur in a ring-shaped region with a similar distance as the real target. Accordingly, Griebel et al. [199] designed a ring grouping to replace the multi-scale grouping in PointNet++. The scene structure and relationship between detections are important cues to identify ghost objects. Garcia et al. [200] suggest the occupancy grid map can provide the information of the scene structure. Therefore, they used the occupancy grid map and the list of moving objects as inputs to FCN, to predict a heat map of moving ghost detections. Wang et al. [201] proposed to use multimodal transformers to capture the semantic affinity between ghost objects and real objects with LiDAR as the reference. They designed a multimodal attention block, which consists of two modules. The first one is a self-attention module for the radar point cloud. It is expected to model the similarities of real objects and mirrored ones. The feature maps from the radar and LiDAR branches are then fused by a second multimodal attention module. This fusion module can be seen as calculating the correlation between LiDAR detections and real radar detections.

### 7.2. Uncertainty in Radar Detection

Learning-based radar detection shows its potential in classifying different road users. However, the performance evaluated on research datasets could be biased due to class imbalance and simple scenarios. Palffy et al. [50] summarised some failure cases for radar detection in the VoD dataset: Two close pedestrians can be detected as one bicyclist. One large object, for example a truck or a bus, can be split into two smaller ones. Distant objects with few reflections may be missed by the detector. Strong reflections from metal poles and high curbs can mask real objects. Most of these failures come from the imperfection of radar sensors with respect to angular resolution and dynamic range. To make it worse, neural networks tend to be overconfident in their incorrect predictions [202]. For autonomous driving, the misspecified confidence in perception can leak to downstream tasks such as sensor fusion and decision making, potentially leading to catastrophic failure. Patel et al. [203] investigated the class uncertainty of a learning-based radar classifier under different perturbations, including domain shift, signal corruptions, and out-of-distribution data. Experiments indicate their baseline network are severely over-confident under these perturbations.

There are two kinds of uncertainty: Data uncertainty, also known as aleatoric uncertainty, is caused by noisy input. Model uncertainty, also known as epistemic uncertainty, is caused by insufficient or inappropriate training of the network. Sources of model uncertainty include three cases: covariate shift (p(x) changes), label shift (p(y) changes), and open set recognition (unseen *y*) [204]. The sum of data uncertainty and model uncertainty is referred to as predictive uncertainty. For the task of probabilistic object detection [205], uncertainties of two parameters are of interest: class uncertainty, which encodes the confidence in the classification, and spatial uncertainty, which represents the reliability of the bounding boxes. Class uncertainty can be seen as model uncertainty, while spatial uncertainty is more relevant to data uncertainty introduced by noisy input.

For classification tasks, the simplest way is to learn a function that maps the pseudo probability output by the softmax layer into the true probability. The true probability is defined as the classwise accuracy on the training set. This process is usually called network calibration. Since it is a postprocessing method, both the model size and inference time are not affected. The calibration method is mainly concerned with the aleatoric part of the overall uncertainty [192]. To calibrate the radar classifier, Patel et al. [206] compared different postprocessing techniques, including temperature scaling [202], the latent Gaussian process (GP) [207], and mutual information maximisation [208]. Mutual information maximisation achieves the best balance between performance and inference time. Some recent research indicates that soft-label augmentation techniques, such as label smoothing [209] and mixup [210,211], can effectively mitigate the over-confidence problem, thus helping network calibration. Patel et al. [206] suggest the use of label smoothing regularisation in radar classification. The core idea is that the classifier should give lower confidence to distant objects with low received power. Therefore, they proposed two label smoothing techniques to generate soft labels according to the range and the received power. respectively. Experiments showed that both of them can significantly improve the calibration performance.

In addition to calibrating the class uncertainty, we are also interested in estimating the spatial uncertainty in bounding box regression. Monte Carlo dropout [212] and deep ensembles [213] are popular in estimating predictive uncertainty. However, experiments [214] show that these methods only provide marginal improvements in object detection, but at a high cost. Direct modelling [215] is widely used to estimate the aleatoric uncertainty in bounding box regression. The idea is to let the network estimate both the mean and variance of a prediction. The loss is constructed as
(13)$$L\left(\theta \right)=\frac{1}{N}\sum_{i=1}^{N}\frac{1}{2\sigma {\left(x_{i}\right)}^{2}}{∥y_{i}-f\left(x_{i}\right)∥}^{2}+\frac{1}{2}\log\sigma {\left(x_{i}\right)}^{2}$$
where $\sigma \left(x_{i}\right)$ is the estimated variance, which reduces the penalty with high variance and penalises high variance at the same time. Dong et al. [216] estimated the spatial uncertainty in radar detection using direct modelling. Experiments indicate that adding variance prediction for bounding box parameters can improve detection performance, especially under a high IoU threshold.

### 7.3. Fusion in Adverse Weather

Adverse weather conditions, such as heavy rain, snow, and fog, can be a significant threat to safe driving. Different sensors operate in different electromagnetic wavebands, thus having different robustness to environments. A comparison of the weather effects on different sensors can be found in [217]. Visual perception is susceptible to blur, noise, and brightness distortions [218,219]. In adverse weather, LiDAR suffers from reduced detection range and blocked view in powder snow [220], heavy rain [221], and strong fog [221]. In contrast, radar is more robust under adverse weather. The effect of weather on radar can be divided into attenuation and backscattering [222]. The attenuation effect decreases the received power of the signal, and the backscattering effect increases the interference at the receiver. Experiments [43,223,224,225] reveal attenuation and backscattering under dust, fog, and light rain are negligible for radar, while the performance of radar degrades under heavy rainfall. Zang et al. [222] summarised the mathematical models for the attenuation and backscattering effects of rain. They suggest the detection range of radar can be reduced by up to 45% under severe rainfall conditions (150 mm/h). For close targets with small RCS, the backscattering effect is more severe and can cause additional performance degradation.

Driving in adverse weather can be considered as a corner case [226] for autonomous driving. The concept of the operational design domain (ODD) [227] is proposed to define the conditions under which autonomous vehicles are designed to operate safely. If the monitoring system [228] detects a violation of OOD requirements, control will be handed over to the driver. However, changes in the operational environment are usually rapid and unpredictable. Therefore, the hand-over mechanism is controversial in terms of safety. In the future, a fully autonomous vehicle (SAE Level 5) is expected to work under all environmental and weather conditions. However, most fusion methods are not designed to explicitly consider weather effects. Networks trained in good weather may experience performance degradation in adverse weather.

There are some possible ways to adapt the network to different weather conditions. One way is to add a scene switching module [229], then use different networks for different weather. This method is straightforward, but introduces additional computational and memory costs. The other option is to add some dynamic mechanisms into the network. Qian et al. [33] added a two-stage attention block in the fusion module. They first applied self-attention to each modality and then mixed them through cross attentions. Experiments showed that the fusion mechanism performs robustly in foggy weather. They further investigated the domain generalisation problem, i.e., training with a good weather dataset and inference in foggy weather. The result showed a significant accuracy drop compared with training with data in both good and foggy weather, indicating that their model relies on data to generalise. Malawade et al. [230] proposed a gating strategy to rank each modality and pick the top three reliable modalities for fusion. They compared three types of gating methods: knowledge-based, CNN-based, and attention-based gating. Knowledge-based gating uses a set of pre-defined modalitywise weights for each weather condition, while CNN-based and attention-based learn the weights from data. Experiments on the RADIATE dataset [44] indicate gating methods outperform fusion methods under adverse weather, and attention-based gating can achieve the best performance. Alternatively, Bijelic et al. [43] proposed an entropy-steered fusion network, which uses the sensor entropy as modalitywise weights. Specifically, they used a deep fusion architecture that continuously fuses feature maps from different modalities. The pixelwise entropy is used as the attention map for each sensor branch. Since the entropy map is conditioned only on sensor inputs, the fusion network can perform robustly in unseen adverse weather. According to uncertainty theory, sensor entropy can be considered as a measure of data uncertainty. To utilise both data and model uncertainties, Ahuja et al. [231] proposed an uncertainty-aware fusion framework. They leveraged a decision-level fusion architecture and expected each branch to output both data uncertainty and model uncertainty. A gating function was used to apply a weighted average to each modality according to the predicted uncertainty. Then, they designed two modules to handle the data with high uncertainty. One for failure detection. A sensor with data uncertainty consistently above a threshold is considered to be malfunctioning. The other is used for continuous learning. Data with model uncertainty above a threshold will be added to the training set for continuous learning.

## 8. Future Research Directions

In this paper, we summarised the recent developments on deep radar perception. As we can see, many research efforts have focused on developing models for detection tasks. However, there are also some unexplored research topics or fundamental questions to be addressed. In this section, we propose some interesting research directions to the automotive radar community.

### 8.1. High-Quality Datasets

The deep learning revolution started with the introduction of the ImageNet dataset [232]. However, radar perception has not yet seen its ImageNet moment. Although many datasets exist, they differ in scale, resolution, data representation, scenario, and labelling granularity. The granularity and quality of labelling are also key issues for radar datasets. Therefore, it is hard to fairly compare different models trained on different datasets. Since the introduction of 4D imaging radar to the market, we anticipate an urgent need for datasets with high-quality annotations and diverse scenes.

### 8.2. Radar Domain Knowledge

In the absence of high-quality datasets, we need to avoid treating AI in radar as a data fitting game. It is essential to exploit domain knowledge to develop a generalizable perception model. Radar-domain knowledge need to be considered at many stages, such as labelling, data augmentation, model structure, training techniques, and evaluation metrics. Take ghost detection as an example. From a data perspective, we need to use our expert knowledge to label ghost objects [61]. From a network perspective, we can design an attention module [201] or utilise graph convolution [167] to model the relationship between real and ghost objects. We hope that researchers put more focus on solving these critical problems in radar perception.

### 8.3. Uncertainty Quantification

As introduced in Section 7.2, uncertainty quantification is important for applying AI in safety-critical applications. Due to the low SNR of radar data and the small size of radar datasets, both high data and model uncertainties are expected for CNN-based radar detectors. However, there is still very little work touching on this topic. Although many uncertainty quantification methods have been proposed, they are not necessarily helpful for a specific task. For example, Feng et al. [233] found that sampling-based methods are not useful for visual object detection. Similarly, we need empirical experiments and theoretical explanations to demonstrate the necessity and effectiveness of uncertainty quantisation methods for radar perception.

### 8.4. Motion Forecasting

An overlooked feature of radar is Doppler velocity. In addition to being a feature of moving road users, Doppler velocity is valuable for motion forecasting. Motion forecasting is a popular research topic in autonomous driving [234]. By accurately estimating the motion of road users, the down-stream path planning module can better react to future interactions. Lin et al. [235] predicted trajectories by building a constant velocity model with binarised RA maps as input. However, experiments showed that the constant velocity model performs poorly in predicting vehicle trajectories [234]. As mentioned in Section 4, a second-order nonlinear motion model can be developed using the measured Doppler velocity. We believe that radar has great potential to play an important role in motion forecasting.

### 8.5. Interference Mitigation

For FMCW radar, mutual interference is a challenging task to solve. It occurs when multiple radars operate simultaneously in a direct line of sight [236]. Depending on if the chirp configuration, i.e., slope and chirp duration, is the same between interferer and victim radar, interference can be classified as coherent and incoherent [237]. Coherent interference occurs when the same chirp configuration is used and leads to ghost detections. Incoherent interference is caused by different types of chirps, resulting in a significantly increased noise floor, masked weak target, and thus, reduced probability of detection. In reality, partially coherent interference is more widely seen where the interferer has a slightly different chirp configuration. Oyedare et al. [238] summarised deep learning methods for interference mitigation. Although these methods achieve better performance than classical zeroing methods, they are generally designed for specific types of disturbances and require significant computational costs. Future research should consider interference mitigation and downstream tasks (e.g., detection) as a whole and build an end-to-end learning framework to optimise them together.

## 9. Conclusions

The purpose of this review article is to provide a big picture of deep radar perception. We first summarised the principles of radar signal processing. Then, we presented a detailed summary of radar datasets for autonomous driving. To encourage researchers to build their own datasets, we also presented methods for calibration and labelling. We further investigated data augmentation and synthetic radar data to improve data diversity. Radar can be used for depth completion and velocity estimation. For the ease of depth completion, several expansion methods were introduced to better associate the radar detections with the image pixels. The full velocity can be recovered as a geometric optimisation problem or in a self-supervised learning way.

Radar detection was the main focus of this paper. We classified deep radar detectors into point-cloud-based and pre-CFAR-based. PointNet variant networks and multi-view encoder–decoders are popular choices for radar point clouds and radar tensors, respectively. By increasing spatial density and exploiting temporal information, significant performance improvements can be achieved. Some new operators, such as depthwise convolution, attention, and graph convolution, are leveraged for larger receptive field. In practical applications, radar is often fused with cameras and LiDAR. We classified fusion frameworks into four categories. Input fusion requires a lightweight preprocessing to explicitly handle radar position imprecision. Cascaded ROI fusion is not robust to sensor failures, while parallel ROI fusion improves it. Feature map fusion provides the network with greater flexibility to combine radar and visual semantics, but requires specific training techniques for effective learning. Decision fusion takes advantage of modal redundancy and is therefore popular in real-world applications. Location information can be robustly fused in a track-to-track architecture or with the help of network semantics. Category information can be fused with Bayesian inference or evidence theory.

We summarised three challenges for deep radar perception. Firstly, multi-path effects need to be explicitly considered in object detection. Secondly, we need to alleviate the problem of overconfidence in radar classification and estimate the uncertainty in bounding box regression. Thirdly, the fusion architecture should have adaptive mechanisms to take full advantage of radar’s all-weather capabilities. Finally, some future research directions were proposed. There is an urgent need for high-quality radar datasets. Radar domain knowledge and uncertainty quantification can help us to develop a generalizable AI model. Considering the perceptual system as a whole, we can extend the end-to-end learning framework forward, i.e., joint learning with interference mitigation, or backward, i.e., predicting motion.

## Figures and Tables

**Figure 1 sensors-22-04208-f001:**
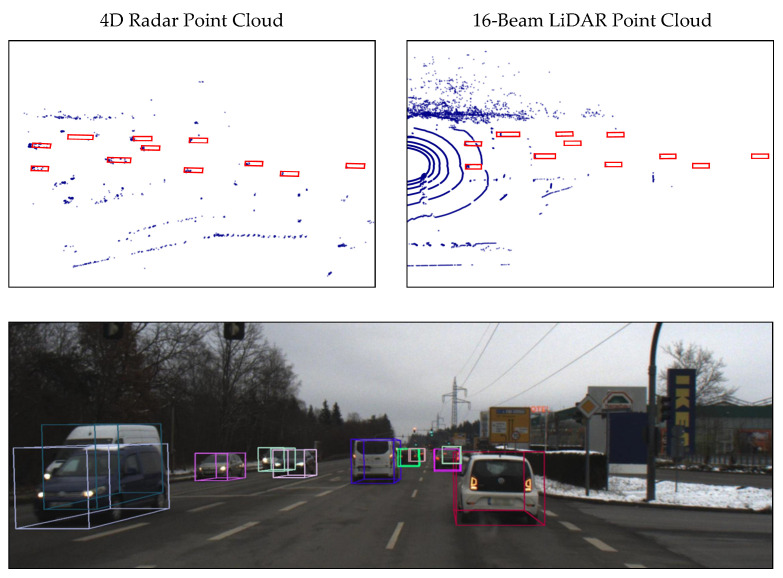
Point clouds of a 4D radar and a 16-beam LiDAR from the Astyx dataset [2].

**Figure 2 sensors-22-04208-f002:**
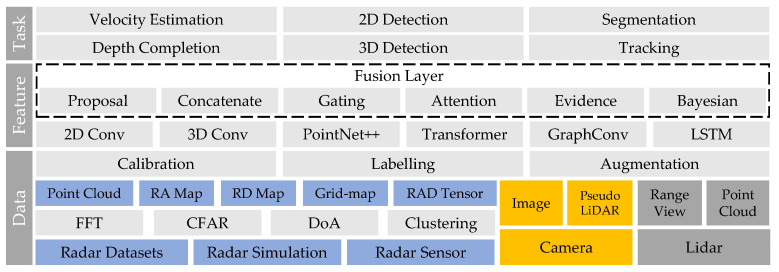
Overview of deep radar perception framework.

**Figure 3 sensors-22-04208-f003:**
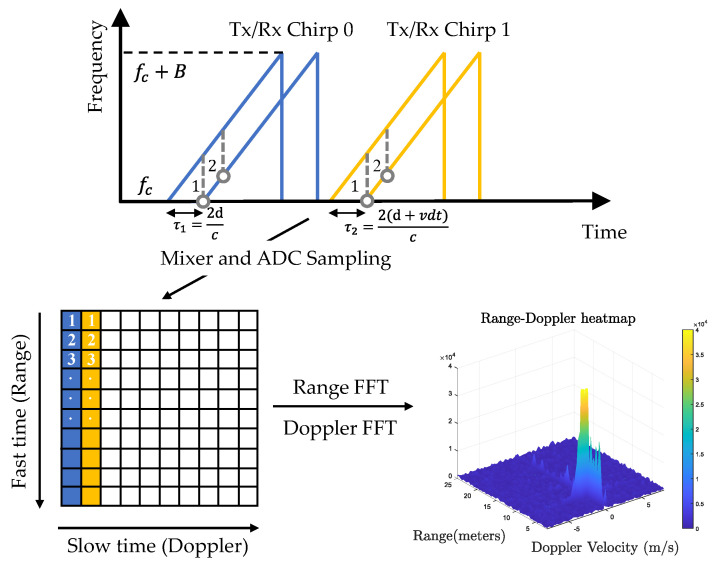
Radar Tx/Rx signals and the resulting range-Doppler map.

**Figure 4 sensors-22-04208-f004:**
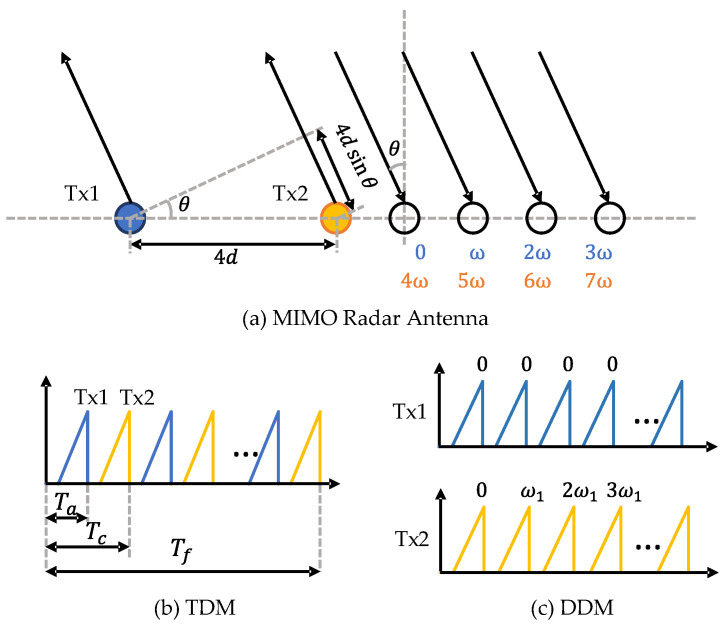
MIMO radar principles. (**a**) Virtual array configuration of a 2Tx4Rx MIMO radar (**b**) In TDM mode, Tx1 and Tx2 transmit signals in turns. (**c**) In DDM mode, a Doppler shift is added to Tx2.

**Figure 5 sensors-22-04208-f005:**
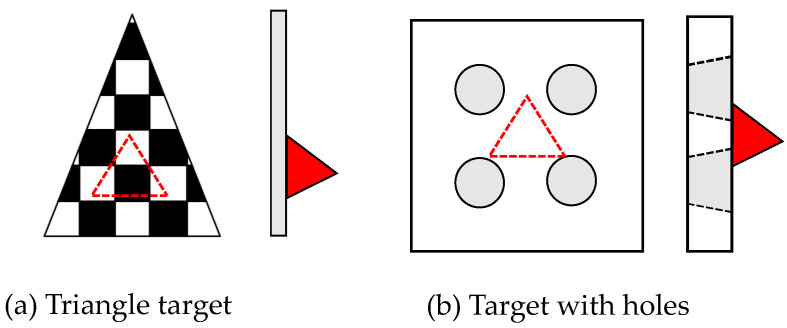
Two types of radar calibration targets [64,65]. The front board is made of styrofoam. The red triangle is a radar corner reflector.

**Figure 6 sensors-22-04208-f006:**
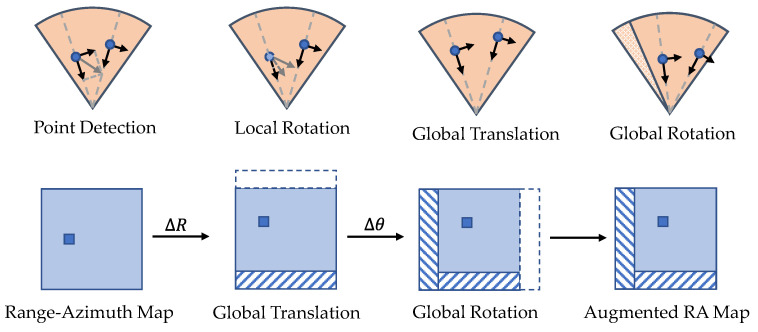
Radar data augmentation techniques. The Doppler velocity measured by radar is a scalar, so local rotation of the radar detection will cause a misalignment between the Doppler velocity and the true velocity. Global translation and rotation are free from such misalignment. When augmenting the radar RA map, it is necessary to interpolate the background area and compensate the intensity of detection.

**Figure 7 sensors-22-04208-f007:**
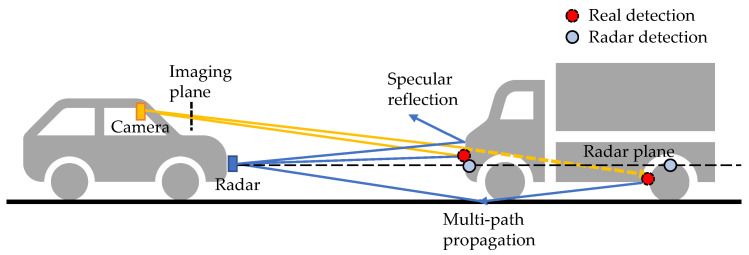
Radar range measurements. Off-the-shelf radars return detections on a 2D radar plane. The detections are sparsely spread on objects due to specular reflection. Due to multi-path propagation, radar can see through occlusions, and meanwhile, this can cause some noisy detections.

**Figure 8 sensors-22-04208-f008:**
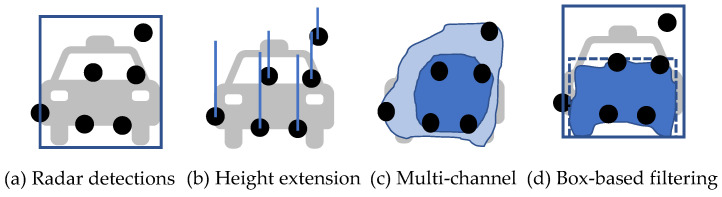
(**a**) Radar detection expansion techniques. (**b**) Extend radar detections in height. (**c**) Build a probabilistic map, where the dark/light blue indicates channel with high/low confidence threshold. (**d**) Apply a strict filtering according to the bounding box, where only detections corresponding to the frontal surface are retained.

**Figure 9 sensors-22-04208-f009:**
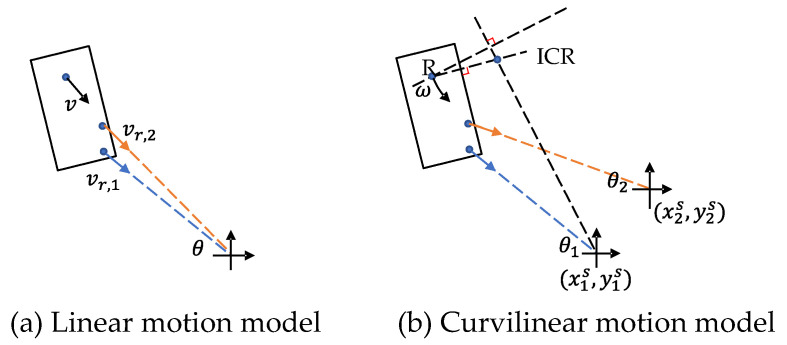
Radar motion model. (**a**) Linear motion model needs multiple detections for the object. (**b**) Curvilinear motion model requires either two radars to observe the same objects or the determination of the vehicle boundary and rear axle.

**Figure 10 sensors-22-04208-f010:**
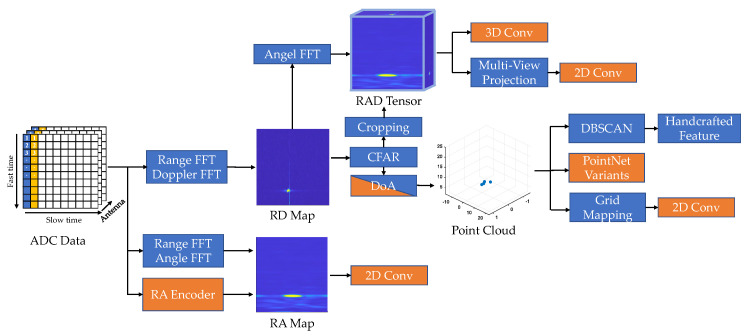
Overview of radar detection frameworks: Blue boxes indicate classical radar detection modules. Orange boxes represent AI-based substitutions.

**Figure 11 sensors-22-04208-f011:**
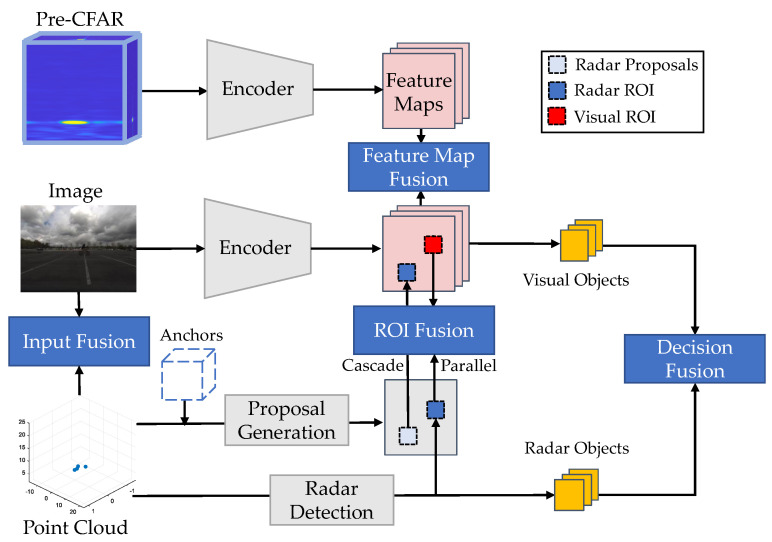
Overview of radar and camera fusion frameworks. We classify the fusion frameworks into input fusion, ROI fusion, feature map fusion, and decision fusion. For ROI fusion, we further investigate two architectures: cascade fusion, which projects radar proposals to image view, and parallel fusion, which fuses radar ROIs and visual ROIs.

**Figure 12 sensors-22-04208-f012:**
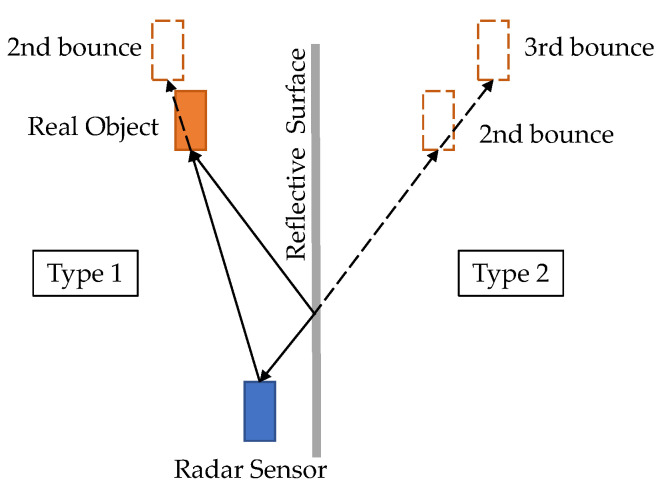
Multi-path effect: The solid orange box is the real object. Dotted boxes are ghost objects caused by multi-path propagation.

**Table 1 sensors-22-04208-t001:** Characteristics of typical radars and LiDARs.

	Conventional Radar ^1^ (Multi-Mode)	4D Radar	16-Beam LiDAR	32-Beam LiDAR	Solid State LiDAR
Max Range	f: 250 m, n: 70 m	300 m	100 m	200 m	200 m
FoV (H/V)	f: 20°, n: 120°/✗	120°/ 30°	360°/30°	360°/40°	120° 25°
Ang Res (H/V)	f: 1.5°, n: 4°/✗	1°/1°	0.1°/2°	0.1°/0.3°	0.2°/0.2°
Doppler Res	0.1 m/s	0.1 m/s	✗	✗	✗
Point Density	Low	Medium	High	High	High
All Weather	✓	✓	✗	✗	✗
Power	5 W	5 W	8 W	10 W	15 W
Expected Cost	Low	Low	Medium	High	Medium

^1^ A typical 77 GHz 4Tx-6Rx automotive radar, 2Tx-6Rx for far range, and 2Tx-6Rx for near range.

**Table 2 sensors-22-04208-t002:** Equations for radar performance.

Definition	Equation
Max Unambiguous Range	$R_{m}=\frac{cB_{\mathrm{IF}}}{2S}$
Max Unambiguous Velocity	$v_{m}=\frac{\lambda }{4T_{c}}$
Max Unambiguous Angle	$\theta_{\mathrm{FoV}}=\pm \arcsin\left(\frac{\lambda }{2d}\right)$
Range Resolution	$\Delta R=\frac{c}{2B}$
Velocity Resolution	$\Delta v=\frac{\lambda }{2N_{c}T_{c}}$
Angular Resolution	$\Delta \theta_{\mathrm{res}}=\frac{\lambda }{N_{R}dcos\left(\theta \right)}$
3 dB Beamwidth	$\Delta \theta_{3\mathrm{dB}}=2\arcsin\frac{1.4\lambda }{\pi D}$

The meaning of parameters is consistent in this section. Refer to Appendix A (in Appendix A) for a quick check of the
meaning.

**Table 3 sensors-22-04208-t003:** Typical automotive radar parameters [17].

Parameter	Range
Transit power (dBm)	10–13
TX/RX antenna gain (dBi)	10–25
Receiver noise figure (dB)	10–20
Target RCS (dBsm)	(−10)–20
Receiver sensitivity (dBm)	(−120)–(−115)
Minimum SNR (dB)	10–20

**Table 4 sensors-22-04208-t004:** Radar datasets.

Name	Year	Task	Radar Type	Data	Doppler	Range	Other Sensors	Scenarios	Weather	Annotations	Size
**Radar Datasets for Detection**											
nuScenes [34]	2020	DT	LR	PC	✓	SV	CLO	USH	✓	3D, T	L
PixSet [35]	2021	DT	LR	PC	✓	MR	CLO	USP	✓	3D, T	M
RadarScenes [36]	2021	DTS	HR	PC	✓	SV	CO	USHT	✓	P${}_{w}$	L
Pointillism [37]	2020	D	2LR	PC	✓	MR	CL	U	✓	3D	M
Zendar [38]	2020	D	SAR	ADC, RD, PC	✓	MR	CLO	U	✗	P${}_{w}$	S
Dense [43]	2020	D	LR	PC	✓	LR	CLO	USHT	✓	3D	L
RADIATE [44]	2020	LDT	SP	RA	✗	SV	CLO	USHP	✓	2D, T, P${}_{s}$	M
**Radar Pre-CFAR Datasets for Detection**											
CARRADA [45]	2020	DTS	LR	RAD	✓	SR	C	R	✗	2D, P${}_{w}$, M, T	M
RADDet [46]	2021	D	LR	RAD	✓	SR	C	US	✗	2D	M
CRUW [47]	2021	D	LR	RAD	✓	USR	C	USHP	✗	P${}_{o}$	L
RaDICaL [48]	2021	L	LR	ADC	✓	USR, SR	C${}_{d}$O	USHIP	✗	2D	L
Ghent VRU [49]	2020	DS	LR	RAD	✓	SR	CL	U	✗	M	M
**4D Radar Datasets for Detection**											
Astyx [2]	2019	D	HR	PC	✓	MR	CL	SH	✗	3D	S
View-of-Delft [50]	2022	DT	HR	PC	✓	SR	CLO	U	✗	3D, T	S
RADIal [51]	2021	DS	HR	ADC, RAD, PC	✓	MR	CLO	USH	✗	P${}_{o}$, M	M
TJ4DRadSet [52]	2022	DT	HR	PC	✓	LR	CLO	U	✗	3D, T	M
**Radar Datasets for Localisation**											
Oxford [53]	2020	L	SP	RA	✗	SV	CLO	U	✓	P${}_{s}$	L
Mulran [54]	2020	L	SP	RA	✗	SV	LO	US	✗	P${}_{s}$	M
Boreas [55]	2022	LD	SP	RA	✗	SV	CLO	S	✓	P${}_{s}$, 3D	L
EU Long-term [56]	2020	L	LR	PC	✓	LR	CLO	U	✓	P${}_{s}$	M
Endeavour [57]	2021	L	LR	PC	✓	5LR	LO	S	✗	P${}_{s}$	M
ColoRadar [58]	2021	L	HR, LR	ADC, PC	✓	2USR	LO	SIT	✓	P${}_{s}$	M
**Radar Datasets for Other Tasks**											
PREVENTION [59]	2019	DT	LR	PC	✓	1LR, 2SR	CLO	UH	✓	2D, T	L
SCORP [60]	2020	S	LR	ADC, RAD	✓	USR	C	P	✗	M	S
Ghost [61]	2021	DS	LR	PC	✗	LR	CLO	S	✗	P${}_{w}$	M

Task: D, T, L, and S stand for detection, tracking, localisation, and segmentation; Type: LR, HR, SP, and SAR stand for low-resolution, high-resolution, spinning, and SAR; Range: SV, LR, MR, SR, an USR stand for surrounding view, long-range (<250 m), middle-range (<180 m), short-range (<50 m), and ultra-short-range (<25 m); Other Sensors: C, C_*d*_, L, and O stand for camera, RGBD camera, LiDAR, and odometry; Scenarios: U, S, H, P, T, R, and I stand for urban (city), suburban, highway, parking lot, tunnel, race track, and indoors; Size: L, M, and S stand for large, medium, and small;Weather stands for adverse weather; Label: 2D, 3D, T, P_*w*_, P_*o*_, P_*s*_, and M stand for 2D bounding box, 3D bounding box, track ID, pointwise detection, object-level point, pose, and segmentation mask.

## Data Availability

Not applicable.

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
