# Peer review of "Towards Deep Radar Perception for Autonomous Driving: Datasets, Methods, and Challenges"

_sensors, 2022, doi:10.3390/s22114208_

Round 1

Reviewer 1 Report

Congratulations on the extensive work with impact and contribution.

Can you show how is the impact of multiple MIMO Radar?

What is the operational mode when we have a Radar interference in an opposed direction?

Could you explain better about the impact of the weather?

Reviewer 2 Report

The paper “Towards Deep Radar Perception for Autonomous Driving: Datasets, Methods, and Challenges” is reviewing recent developments of automotive radars and the challenges they are facing. The paper relates to various features of automotive radar operation, including physical aspects, link behavior, sensing, signal processing and algorithms.

I found the review quiet comprehensive, however as a review paper, I would expect to find definitions of synonyms and terms used along this (long !) manuscript. This includes terms that may be obvious to the readers but have to be defined, such as: Lidar, 4D radars, TDM, DDM.

ALL the symbols used in the formulas MUST be defined.

In such a comprehensive presentation, I expect to see discussion on the critical design considerations. Such a review should be accompanied with some fundamental formulation of the different radar parameters and modulation techniques and their effects on the radar requirements and performances:

Transmitted power requirements

Bandwidth

Range accuracy and resolution

Velocity accuracy and resolution

The effect of signal to noise ratio on the detection performances

Minimum detectible signal (MDS, Pmin) power

Antenna directivity and gain

It will also be important to present some discussion on the propagation of millimeter waves for relevant scenarios, considering weather conditions and multipath.

I think that inclusion of the above will give a wide insight on the development of radars for autonomous vehicles.

Reviewer 3 Report

This review provides a big picture of deep radar perception, introducing datasets and methods for calibration and labelling. English is ok, and the references adequate.

I have just some small remarks:

- In the text, missed links to Tables, Sections, Equations, Figures must be corrected.

- In Table 1, the classification of radars is too general. The characteristics of the radars strongly depend on the operational parameters, so it is necessary to provide more details about the radars considered.

The paper can be accepted with minor revision.

Round 2

Reviewer 2 Report

Note to write Ghz and not GHz.